# An Exploration of Left-Corner Transformations

**Andreas Opedal**[1,2,*]    **Eleftheria Tsipidi**[1,*]    **Tiago Pimentel**[1,3]
**Ryan Cotterell**[1]    **Tim Vieira**[1]

[1]ETH Zürich    [2]Max Planck ETH Center for Learning Systems    [3]University of Cambridge
{andreas.opedal,eleftheria.tsipidi,ryan.cotterell}@inf.ethz.ch
tp472@cam.ac.uk    tim.f.vieira@gmail.com

## Abstract

The left-corner transformation (Rosenkrantz and Lewis, 1970) is used to remove left recursion from context-free grammars, which is an important step towards making the grammar parsable top-down with simple techniques. This paper generalizes prior left-corner transformations to support semiring-weighted production rules and to provide finer-grained control over which left corners may be moved. Our generalized left-corner transformation (GLCT) arose from unifying the left-corner transformation and speculation transformation (Eisner and Blatz, 2007), originally for logic programming. Our new transformation and speculation define equivalent weighted languages. Yet, their derivation trees are structurally different in an important way: GLCT replaces left recursion with right recursion, and speculation does not. We also provide several technical results regarding the formal relationships between the outputs of GLCT, speculation, and the original grammar. Lastly, we empirically investigate the efficiency of GLCT for left-recursion elimination from grammars of nine languages.

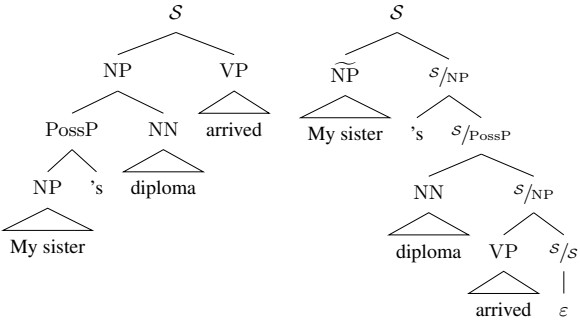
https://github.com/rycolab/left-corner

## 1 Introduction

Grammar transformations are functions that map one context-free grammar to another. The formal language theory literature contains numerous examples of such transformations, including nullary rule removal, rule binarization, and conversion to normal forms, e.g., those of Chomsky (1959) and Greibach (1965). In this work, we study and generalize the left-corner transformation (Rosenkrantz and Lewis, 1970). Qualitatively, this transformation maps the derivation trees of an original grammar into isomorphic trees in the transformed grammar. The trees of the transformed grammar will be such that the base subtree of a left-recursive chain (the left corner) is *hoisted* up in a derivation tree while replacing the left-recursive path to the

---

[*]Equal contribution.

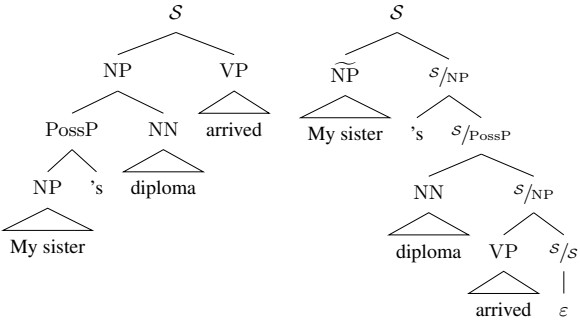

Figure 1: *Left:* An example derivation containing left-recursive rules. *Right:* The corresponding derivation after our left-corner transformation. Observe that (i) the lower NP subtree has been hoisted up the tree, (ii) the *left* recursion from NP to NP has been replaced with *right* recursion from $\mathcal{S}/\text{NP}$ to $\mathcal{S}/\text{NP}$.

left corner with a right-recursive path to an empty constituent. Fig. 1 provides an example.

A common use case of the left-corner transformation is to remove left recursion from a grammar, which is necessary for converting the grammar to Greibach normal form and for several top-down parsing algorithms (Aho and Ullman, 1972). As an additional effect, it reduces the stack depth of top-down parsing (Johnson, 1998), which makes it an interesting method for psycholinguistic applications (Roark et al., 2009; Charniak, 2010). The closely related left-corner *parsing* strategy has been argued to be more cognitively plausible than alternatives due to its constant memory load across both left- and right-branching structures and low degree of local ambiguity for several languages (Johnson-Laird, 1983; Abney and Johnson, 1991; Resnik, 1992); indeed, empirical evidence has shown that certain left-corner parsing steps correlate with brain activity (Brennan and Pylkkänen, 2017; Nelson et al., 2017) and reading times (Oh et al., 2022).[1]

This paper uncovers an interesting connection between the left-corner transformation and

---

[1]Moreover, statistical left-corner parsers have proven themselves empirically effective for several grammar formalisms (Roark and Johnson, 1999; Roark, 2001; Díaz et al., 2002; Noji and Miyao, 2014; Noji et al., 2016; Shain et al., 2016; Stanojević and Stabler, 2018; Kitaev and Klein, 2020).

the speculation transformation (Eisner and Blatz, 2007).[2] We show that speculation also hoists subtrees up the derivation tree, as in the left-corner transformation. However, in contrast, it does *not* remove left recursion. In uncovering the similarity to speculation, we discover that speculation has been formulated with more specificity than prior left-corner transformations: it has parameters that allow it to control which types of subtrees are permitted to be hoisted and which paths they may be hoisted along. We bring this flexibility to the left-corner transformation with a novel *generalized left-corner transformation* (GLCT).[3] It turns out that the latter functionality is provided by the selective left-corner transformation (Johnson and Roark, 2000); however, the former is new.

We provide several new technical results:[4] (i) We prove that GLCT preserves the weighted languages and that an isomorphism between derivation trees exists (Theorem 1). (ii) We provide explicit methods for mapping back and forth between the derivation trees (Algs. 1 and 2). (iii) We prove that the set of derivation trees for speculation and GLCT are isomorphic (Theorem 2). (iv) We prove that our GLCT-based left-recursion elimination strategy removes left recursion (Theorem 4). Additionally, we empirically investigate the efficiency of GLCT for left-recursion elimination from grammars of nine different languages in §4.1.

## 2 Preliminaries

This section provides the necessary background on the concepts pertaining to semiring-weighted context-free grammars that this paper requires.

**Definition 1.** *A **semiring** is a tuple $\langle \mathbb{W}, \oplus, \otimes, \mathbf{0}, \mathbf{1} \rangle$ where $\mathbb{W}$ is a set and the following hold:*

- *$\oplus$ is an associative and commutative binary operator with an identity element $\mathbf{0} \in \mathbb{W}$*
- *$\otimes$ is an associative binary operator with an identity element $\mathbf{1} \in \mathbb{W}$*
- *Distributivity: $\forall a, b, c \in \mathbb{W}, (a \oplus b) \otimes c = (a \otimes c) \oplus (b \otimes c)$ and $c \otimes (a \oplus b) = (c \otimes a) \oplus (c \otimes b)$*
- *Annihilation: $\forall a \in \mathbb{W}, a \otimes \mathbf{0} = \mathbf{0} \otimes a = \mathbf{0}$*

*The semiring is commutative if $\otimes$ is commutative.*

We highlight a few commutative semirings and their use cases: • boolean $\langle \{\bot, \top\}, \vee, \wedge, \bot, \top \rangle$: string membership in a language, • nonnegative real $\langle \mathbb{R}_{\geq 0} \cup \{\infty\}, +, \cdot, 0, 1 \rangle$: the total probability of a string, • viterbi $\langle [0, 1], \max, \cdot, 1, 0 \rangle$: the weight of the most likely derivation of a string. For further reading on semirings in NLP, we recommend Goodman (1999) and Huang (2008).

Our work studies weighted context-free grammars (WCFGs), which define a tractable family of weighted languages—called weighted context-free languages—that is frequently used in NLP applications (see, e.g., Jurafsky and Martin, 2020).

**Definition 2.** *A **weighted context-free grammar** is a tuple $\mathfrak{G} = \langle \mathcal{N}, \mathcal{V}, \mathcal{S}, \mathcal{R} \rangle$ where*

- *$\mathcal{N}$ is a set of nonterminal symbols*
- *$\mathcal{V}$ is a set of terminal symbols, $\mathcal{V} \cap \mathcal{N} = \emptyset$*
- *$\mathcal{S} \in \mathcal{N}$ is the start symbol*
- *$\mathcal{R}$ is a bag (multiset) of weighted production rules. Each rule $r \in \mathcal{R}$ is of the form $\mathrm{X} \xrightarrow{w} \boldsymbol{\alpha}$ where $\mathrm{X} \in \mathcal{N}$, $\boldsymbol{\alpha} \in (\mathcal{V} \cup \mathcal{N})^*$, and $w \in \mathbb{W}$. We assume that $\mathbb{W}$ is commutative semiring.*

We will use the following notational conventions: • $\mathrm{X}, \mathrm{Y}, \mathrm{Z} \in \mathcal{N}$ for nonterminals • $a, b, c \in \mathcal{V}$ for terminals • $\mathbf{x}, \mathbf{y}, \mathbf{z} \in \mathcal{V}^*$ for a sequence of terminals • $\alpha, \beta, \gamma \in (\mathcal{V} \cup \mathcal{N})$ for a terminal or nonterminal symbol • $\boldsymbol{\alpha}, \boldsymbol{\beta}, \boldsymbol{\gamma} \in (\mathcal{V} \cup \mathcal{N})^*$ for a sequence of nonterminals or terminals • $\mathfrak{G} = \langle \mathcal{N}, \mathcal{V}, \mathcal{S}, \mathcal{R} \rangle$ and $\mathfrak{G}' = \langle \mathcal{N}', \mathcal{V}', \mathcal{S}', \mathcal{R}' \rangle$ for WCFGs. We write $w(r)$ to access the weight of a rule $r$.

When describing a grammar and its rules, we may use the following terms: The **size** of a grammar is $\sum_{(\mathrm{X} \to \boldsymbol{\alpha}) \in \mathcal{R}} (1 + |\boldsymbol{\alpha}|)$. The **arity** of a rule $\mathrm{X} \to \boldsymbol{\alpha}$ is equal to $|\boldsymbol{\alpha}|$; thus, a rule is **nullary** if $|\boldsymbol{\alpha}| = 0$, **unary** if $|\boldsymbol{\alpha}| = 1$, and so on. For technical reasons, it is often convenient to eliminate nullary rules from the grammar.[5]

A **derivation** is a rooted, $(\mathcal{N} \cup \mathcal{V})$-labeled, ordered tree where each internal node must connect to its children by a production rule.[6] To access the root **label** of a derivation $\boldsymbol{\delta}$, we write $\ell(\boldsymbol{\delta})$. The **set of all derivations** of $\mathfrak{G}$ is the smallest set $\mathcal{D}$

---

[2]Speculation was originally a transformation for weighted logic programs, which we have adapted to CFGs.

[3]We note that generalized left-corner *parsers* also exist (Demers, 1977; Nederhof, 1993). However, they are generalized differently to our transformation.

[4]These technical results fill some important gaps in the literature, as prior work (Rosenkrantz and Lewis, 1970; Johnson, 1998; Johnson and Roark, 2000; Moore, 2000; Eisner and Blatz, 2007) did not provide formal proofs.

[5]Opedal et al. (2023, §F) provides an efficient method to remove nullary rules from semiring-weighted CFGs, which we make use of in our experiments (§4.1).

[6]Note a derivation may be built by a rule with an empty right-hand side; thus, the leaves may be elements of $\mathcal{N}$. When rendering such a derivation, childlessness is marked with $\varepsilon$.

satisfying

$$\mathcal{D} = \mathcal{V} \cup \tag{1}$$

$$\left\{ \begin{array}{c} \overset{\text{X}}{\overbrace{\alpha_1 \cdots \alpha_K}} \\ \triangle \qquad \triangle \end{array} \,\middle|\, \begin{array}{l} (\text{X} \to \alpha_1 \cdots \alpha_K) \in \mathcal{R}, \\ \underset{\triangle}{\alpha_1} \in \mathcal{D}, \dots, \underset{\triangle}{\alpha_K} \in \mathcal{D} \end{array} \right\}$$

The **yield** $\sigma(\boldsymbol{\delta}) \in \mathcal{V}^*$ of a derivation $\boldsymbol{\delta}$ is

- **if** $\boldsymbol{\delta} \in \mathcal{V}$: $\sigma(\boldsymbol{\delta}) \overset{\text{def}}{=} \ell(\boldsymbol{\delta}) = a$ (2)

- **else:** $\sigma\left( \begin{array}{c} \overset{\text{X}}{\overbrace{\alpha_1 \cdots \alpha_K}} \\ \triangle \quad \triangle \end{array} \right) \overset{\text{def}}{=} \sigma\left( \underset{\triangle}{\alpha_1} \right) \circ \cdots \circ \sigma\left( \underset{\triangle}{\alpha_K} \right)$

where $\circ$ denotes concatenation. The **weight** $\omega(\boldsymbol{\delta}) \in \mathbb{W}$ of a derivation $\boldsymbol{\delta}$ is

- **if** $\boldsymbol{\delta} \in \mathcal{V}$: $\omega(\boldsymbol{\delta}) \overset{\text{def}}{=} \mathbf{1}$ (3)

- **else:** $\omega\left( \begin{array}{c} \overset{\text{X}}{\overbrace{\alpha_1 \cdots \alpha_K}} \\ \triangle \quad \triangle \end{array} \right) \overset{\text{def}}{=} \begin{array}{l} w(\text{X} \to \alpha_1 \cdots \alpha_K) \otimes \\ \omega\left( \underset{\triangle}{\alpha_1} \right) \otimes \cdots \otimes \omega\left( \underset{\triangle}{\alpha_K} \right) \end{array}$

The **weighted language** of $\alpha \in (\mathcal{V} \cup \mathcal{N})$ is a function $\mathfrak{G}_\alpha : \mathcal{V}^* \to \mathbb{W}$ defined as follows:

$$\mathfrak{G}_\alpha(\mathbf{x}) \overset{\text{def}}{=} \bigoplus_{\boldsymbol{\delta} \in \mathcal{D}_\alpha(\mathbf{x})} \omega(\boldsymbol{\delta}) \tag{4}$$

where $\mathcal{D}_\alpha \overset{\text{def}}{=} \{\boldsymbol{\delta} \in \mathcal{D} : \ell(\boldsymbol{\delta}) = \alpha\}$ denotes the subset of $\mathcal{D}$ containing trees labeled $\alpha$, and $\mathcal{D}_\alpha(\mathbf{x}) \overset{\text{def}}{=} \{\boldsymbol{\delta} \in \mathcal{D}_\alpha : \sigma(\boldsymbol{\delta}) = \mathbf{x}\}$ denotes those with yield $\mathbf{x}$. In words, the value of a string $\mathbf{x} \in \mathcal{V}^*$ in the weighted language $\mathfrak{G}_\alpha$ is the $\oplus$-sum of the weights of all trees in $\mathcal{D}_\alpha$ with $\mathbf{x}$ as its yield. The weighted language of the *grammar* $\mathfrak{G}$ is $\mathfrak{G}(\mathbf{x}) \overset{\text{def}}{=} \mathfrak{G}_\mathcal{S}(\mathbf{x})$. Given a set of symbols $\mathcal{X}$, we write $\mathcal{D}_\mathcal{X}$ as shorthand for $\bigcup_{\alpha \in \mathcal{X}} \mathcal{D}_\alpha$. Lastly, let $[\![\boldsymbol{\delta}]\!]$ denote the weighted language generated by the tree $\boldsymbol{\delta}$.[7]

We define the following operations on weighted languages $\mathfrak{G}$ and $\mathfrak{G}'$, for $\mathbf{x} \in \mathcal{V} \cup \mathcal{V}'$:

- Union: $[\mathfrak{G} \oplus \mathfrak{G}'](\mathbf{x}) \overset{\text{def}}{=} \mathfrak{G}(\mathbf{x}) \oplus \mathfrak{G}'(\mathbf{x})$
- Concatenation: $[\mathfrak{G} \circ \mathfrak{G}'](\mathbf{x}) \overset{\text{def}}{=} \bigoplus_{\mathbf{y} \circ \mathbf{z} = \mathbf{x}} \mathfrak{G}(\mathbf{y}) \circ \mathfrak{G}'(\mathbf{z})$

Note that these operations form a (noncommutative) semiring over weighted languages where $\mathbf{0}$ is the language that assigns weight zero to all strings

---

[7]Formally, $[\![\boldsymbol{\delta}]\!](\mathbf{x}) \overset{\text{def}}{=} \omega(\boldsymbol{\delta})$ if $\mathbf{x} = \sigma(\boldsymbol{\delta})$ else $\mathbf{0}$.

and $\mathbf{1}$ is the language that assigns one to the empty string and zero to other strings.

The weighted language of $\mathfrak{G}$ may also be expressed as a certain solution[8] to the following system of equations:

$$\mathfrak{G}_a = \mathbf{1}_a \qquad\qquad \forall a \in \mathcal{V} \tag{5a}$$

$$\mathfrak{G}_\text{X} = \bigoplus_{(\text{X} \overset{w}{\to} \beta_1 \cdots \beta_K) \in \mathcal{R}} w \circ \mathfrak{G}_{\beta_1} \circ \cdots \circ \mathfrak{G}_{\beta_K} \quad \forall \text{X} \in \mathcal{N} \tag{5b}$$

where $\mathbf{1}_a$ is the weighted language that assigns $\mathbf{1}$ to the string $a$ and $\mathbf{0}$ to other strings.

We say that $\alpha$ is **useless** if there does not exist a derivation $\boldsymbol{\delta} \in \mathcal{D}_\mathcal{S}$ that has a subderivation $\boldsymbol{\delta}'$ with $\ell(\boldsymbol{\delta}') = \alpha$. We define **trimming** TRIM($\mathfrak{G}$) as removing each useless nonterminal and any rule in which they participate. It is easy to see that trimming does not change the weighted language of the grammar because no useless nonterminals participate in a derivation rooted at $\mathcal{S}$. We can trim useless rules in linear time using well-known algorithms (Hopcroft and Ullman, 1979).

We say that grammars $\mathfrak{G}$ and $\mathfrak{G}'$ are **equal** ($\mathfrak{G} = \mathfrak{G}'$) if they have the same tuple representation after trimming. We say they are **equivalent** ($\mathfrak{G} \equiv \mathfrak{G}'$) if they define the same weighted language.[9] We say that they are $\mathcal{X}$**-bijectively equivalent** ($\mathfrak{G} \equiv_\mathcal{X} \mathfrak{G}'$) if a structure-preserving bijection of type $\phi : \mathcal{D}_\mathcal{X} \to \mathcal{D}'_\mathcal{X}$ exists. The mapping $\phi$ is **structure-preserving** if $(\forall \boldsymbol{\delta} \in \mathcal{D}_\mathcal{X})$ it is (i) label-preserving ($\ell(\boldsymbol{\delta}) = \ell(\phi(\boldsymbol{\delta}))$), (ii) yield-preserving ($\sigma(\boldsymbol{\delta}) = \sigma(\phi(\boldsymbol{\delta}))$), and (iii) weight-preserving ($\omega(\boldsymbol{\delta}) = \omega(\phi(\boldsymbol{\delta}))$). Suppose $\mathfrak{G} \equiv_\mathcal{X} \mathfrak{G}'$, $\mathcal{S} \in \mathcal{X}$ and $\mathcal{S} = \mathcal{S}'$, then $\mathfrak{G} \equiv \mathfrak{G}'$, but not conversely. A benefit of this stronger notion of equivalence is that derivations in $\mathfrak{G}$ and $\mathfrak{G}'$ are interconvertible: we can parse in $\mathfrak{G}$ and convert to a parse in $\mathfrak{G}'$ and vice versa, assuming $\mathcal{S} = \mathcal{S}'$ and $\mathcal{S} \in \mathcal{X}$.[10]

## 3 Transformations

This section specifies our novel generalized left-corner transformation, its correctness guarantees,

---

[8]Note that the system of equations does not necessarily have a unique solution. In the case of an $\omega$-continuous semiring, (4) coincides with the *smallest* solution to (5) under the natural ordering (Droste and Kuich, 2009).

[9]I.e., $(\mathfrak{G} \equiv \mathfrak{G}') \iff \forall \mathbf{x} \in (\mathcal{V} \cup \mathcal{V}')^* : \mathfrak{G}_\mathcal{S}(\mathbf{x}) = \mathfrak{G}'_{\mathcal{S}'}(\mathbf{x})$.

[10]Under these conditions, our $\mathcal{X}$-bijective equivalence notion becomes a weighted extension of what Gray and Harrison (1969) call *complete covers* and Nijholt (1980) calls *proper covers*. However, their definitions assume traditional string-rewriting derivations instead of tree-structured derivations.

and its connections to prior left-corner transformations. We also describe the speculation transformation (Eisner and Blatz, 2007) and discuss the connections between the speculation transformation and our generalized left-corner transformation.

### 3.1 Generalized Left-Corner Transformation

This section introduces the generalized left-corner transformation (GLCT).[11] This transformation extends prior left-corner transformations by providing additional parameters that control which subtrees can be hoisted.

**Definition 3.** *The **generalized left-corner transformation** $\mathrm{GLCT}(\mathfrak{G}, \overline{\mathcal{R}}, \overline{\mathcal{C}})$ takes as input*

- *a grammar $\mathfrak{G} = \langle \mathcal{N}, \mathcal{V}, \mathcal{S}, \mathcal{R} \rangle$*
- *a subset of (non-nullary) rules $\overline{\mathcal{R}} \subseteq \mathcal{R}$ (called left-corner recognition rules)*
- *a subset of symbols $\overline{\mathcal{C}} \subseteq (\mathcal{V} \cup \mathcal{N})$ (called left-corner recognition symbols)*

*and outputs a grammar $\mathfrak{G}' = \langle \mathcal{N}', \mathcal{V}', \mathcal{S}', \mathcal{R}' \rangle$ with*

- *a superset of nonterminals ($\mathcal{N}' \supseteq \mathcal{N}$)[12]*
- *the same set of terminals ($\mathcal{V}' = \mathcal{V}$)*
- *the same start symbol ($\mathcal{S}' = \mathcal{S}$)*
- *the weighted production rules ($\mathcal{R}'$):*

$$X \xrightarrow{1} \widetilde{X}: \qquad\qquad X \in \mathcal{N} \backslash \overline{\mathcal{C}} \quad (6a)$$

$$X \xrightarrow{1} \widetilde{\alpha}\, X/\alpha: \qquad\qquad X \in \mathcal{N}, \alpha \in \overline{\mathcal{C}} \quad (6b)$$

$$X/X \xrightarrow{1} \varepsilon: \qquad\qquad X \in \mathcal{V} \cup \mathcal{N} \quad (6c)$$

$$Y/\alpha \xrightarrow{w} \boldsymbol{\beta}\, Y/X: \quad X \xrightarrow{w} \alpha \boldsymbol{\beta} \in \overline{\mathcal{R}}, Y \in \mathcal{N} \quad (6d)$$

$$\widetilde{X} \xrightarrow{w} \boldsymbol{\alpha}: \qquad\qquad X \xrightarrow{w} \boldsymbol{\alpha} \in \mathcal{R} \backslash \overline{\mathcal{R}} \quad (6e)$$

$$\widetilde{X} \xrightarrow{w} \widetilde{\alpha}\, \boldsymbol{\beta}: \qquad X \xrightarrow{w} \alpha \boldsymbol{\beta} \in \overline{\mathcal{R}}, \alpha \notin \overline{\mathcal{C}} \quad (6f)$$

*The transformation creates two kinds of new nonterminals using $\widetilde{\alpha}$ and $X/\alpha$.[13]*

In Def. 3, we see that GLCT introduces new nonterminals of two varieties: **slashed** nonterminals (denoted $Y/\alpha$) and **frozen**[14] nonterminals (denoted

---

[11] A generalized *right*-corner transformation can be defined analogously—applications of such a transformation are given in Schuler et al. (2010) and Amini and Cotterell (2022).

[12] $\mathcal{N}' = \mathcal{N} \cup \{\widetilde{X} \mid X \in \mathcal{N}\} \cup \{\alpha/\beta \mid \alpha, \beta \in (\mathcal{N} \cup \mathcal{V})\}$

[13] This notation works as follows: we associate a unique identifier $id$ with the transformation instance. Then, $\widetilde{\alpha} \stackrel{\text{def}}{=} \langle id, \alpha \rangle$ if $\alpha \in \mathcal{N}$ else $\alpha$ and $X/\alpha \stackrel{\text{def}}{=} \langle id, X, \alpha \rangle$. This ensures that the symbols produced cannot conflict with those on $\mathcal{N}$ (i.e., $\forall \alpha \in \mathcal{N} \cup \mathcal{V}, X \in \mathcal{N}: \widetilde{\alpha}, X/\alpha \notin \mathcal{N}$).

[14] Our notion of frozen nonterminals is borrowed directly from the speculation transformation (Eisner and Blatz, 2007), where they are called *other*.

$\widetilde{X}$). The frozen and slashed nonterminals are each defined recursively.

- Slashed nonterminals are built by a base case (6c) and a recursive case (6d).
- Frozen nonterminals are built by a base case (6e) and a recursive case (6f).

We see in (6a) and (6b) that GLCT replaces the rules defining the **original** nonterminals ($\mathcal{N}$) with rules that use GLCT's new nonterminals; the only way to build a nonterminal from $\mathcal{N}$ is using one of these two rules. We refer to these as **recovery rules** because they recover the original symbols from the new frozen and slashed symbols. We also see that (6d) is responsible for converting left recursion into right recursion because the slashed nonterminal on its right-hand side is moved to the right of $\boldsymbol{\beta}$. We will return to this when we discuss speculation in §3.2. Fig. 2 illustrates how GLCT transforms trees.

**Left corner and spine.** To better understand the parameters $\overline{\mathcal{C}}$ and $\overline{\mathcal{R}}$, we define the spine and left corner of a derivation $\boldsymbol{\delta} \in \mathcal{D}$ of the original grammar. Suppose $\boldsymbol{\delta}$ has the following general form:

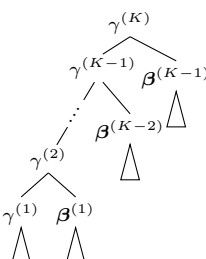

Then, we define the **spine** $\zeta(\boldsymbol{\delta})$ as the maximum-length sequence of rules $(\gamma^{(K)} \to \gamma^{(K-1)} \boldsymbol{\beta}^{(K-1)}) \cdots (\gamma^{(2)} \to \gamma^{(1)} \boldsymbol{\beta}^{(1)})$ along the left edges of $\boldsymbol{\delta}$ where each rule is in $\overline{\mathcal{R}}$. The **left corner** $\lambda(\boldsymbol{\delta})$ of a tree $\boldsymbol{\delta} \in \mathcal{D}$ is the bottommost subtree $\boldsymbol{\delta}_1$ of $\boldsymbol{\delta}$ with $\ell(\boldsymbol{\delta}_1) \in \overline{\mathcal{C}}$ that is reachable starting at the root $\ell(\boldsymbol{\delta})$ along the left edges of $\boldsymbol{\delta}$ where each edge comes from a rule in $\overline{\mathcal{R}}$. If no such subtree exists, we say that $\boldsymbol{\delta}$ has no left corner and write $\lambda(\boldsymbol{\delta}) = \bot$. We write $\boldsymbol{\delta}/\lambda(\boldsymbol{\delta})$ to denote the $\boldsymbol{\delta}$ with $\lambda(\boldsymbol{\delta})$ replaced by the empty subtree; we define $\boldsymbol{\delta}/\bot \stackrel{\text{def}}{=} \boldsymbol{\delta}$. Lastly, we define $D/\alpha \stackrel{\text{def}}{=} \{\boldsymbol{\delta}/\lambda(\boldsymbol{\delta}) \mid \boldsymbol{\delta} \in D, \ell(\lambda(\boldsymbol{\delta})) = \alpha\}$ where $D$ is a set of derivations and $\alpha \in \mathcal{N} \cup \mathcal{V} \cup \{\bot\}$.

To illustrate, let $\boldsymbol{\delta}$ be the derivation on the left in Fig. 1. The right-hand side derivation results from applying GLCT with $\overline{\mathcal{C}} = \{\text{NP}\}$ and $\overline{\mathcal{R}} = \{\mathcal{S} \to \text{NP VP}, \text{NP} \to \text{PossP NN}, \text{PossP} \to \text{NP 's}\}$. The spine of $\boldsymbol{\delta}$, then, is the sequence of rules in $\overline{\mathcal{R}}$ (in the same order), and its left corner is the lower NP-subtree. Note how the left corner is the subtree

hoisted by the transformation.

**Interpretation.** The symbol $^X/_\alpha$ represents the weighted language of X where we have replaced its left corner subtrees labeled $\alpha$ (if one exists) with $\varepsilon$. We can see that the recovery rule (6b) uses these slashed nonterminals to reconstruct X by each of its left-corner types (found in $\overline{\mathcal{C}}$). We also have a recovery rule (6a) that uses a frozen nonterminal $\widetilde{X}$, which represents the other ways to build X (i.e., those that do not have a left corner in $\overline{\mathcal{C}}$). Thus, the weighted language of X decomposes as a certain sum of slashed nonterminals and its frozen version (formalized below).

**Proposition 1** (Decomposition). *Suppose $\mathfrak{G}' = \text{GLCT}(\mathfrak{G}, \overline{\mathcal{R}}, \overline{\mathcal{C}})$. Then, for any $X \in \mathcal{N}$:*

$$\mathfrak{G}_X = \mathfrak{G}'_{\widetilde{X}} \oplus \bigoplus_{\alpha \in \overline{\mathcal{C}}} \mathfrak{G}'_{\widetilde{\alpha}} \circ \mathfrak{G}'_{X/\alpha} \tag{7}$$

See Appendix A for proof. Next, we describe the weighted languages of the slashed and frozen non-terminals in relation to the derivations in the original grammar. Proposition 2 establishes that the weighted language of $\widetilde{\alpha}$ is the total weight of $\alpha$ derivations without a left corner, and $^X/_\alpha$ is the total weight of all X derivations with an $\alpha$ left corner that has been replaced by $\varepsilon$.

**Proposition 2** (Weighted language relationship). *Suppose $\mathfrak{G}' = \text{GLCT}(\mathfrak{G}, \overline{\mathcal{C}}, \overline{\mathcal{R}})$. Then, for any $X \in \mathcal{N}$ and $\alpha \in \mathcal{N} \cup \mathcal{V}$:*

$$\mathfrak{G}'_{\widetilde{\alpha}} = \bigoplus_{\delta \in \mathcal{D}_\alpha/\perp} [\![\delta]\!] \quad and \quad \mathfrak{G}'_{X/\alpha} = \bigoplus_{\delta \in \mathcal{D}_X/\alpha} [\![\delta]\!]$$

See Appendix B for proof.

**Special cases.** We now discuss how our transformation relates to prior left-corner transformations. The **basic left-corner transformation** (Rosenkrantz and Lewis, 1970; Johnson, 1998) is $\text{LCT}(\mathfrak{G}) = \text{GLCT}(\mathfrak{G}, \mathcal{R}, \mathcal{N} \cup \mathcal{V})$, i.e., we set $\overline{\mathcal{R}} = \mathcal{R}$ and $\overline{\mathcal{C}} = \mathcal{N} \cup \mathcal{V}$.[15] This forces the left-most leaf symbol of the tree to be the left corner, which is either a terminal or the left-hand side of a nullary rule. The **selective left-corner transformation** (SLCT; Johnson and Roark, 2000) $\text{SLCT}(\mathfrak{G}, \overline{\mathcal{R}}) = \text{GLCT}(\mathfrak{G}, \overline{\mathcal{R}}, \mathcal{N} \cup \mathcal{V})$ supports left-corner recognition rules $\overline{\mathcal{R}}$, but it does not allow

control over the left-corner recognition symbols, as $\overline{\mathcal{C}}$ is required to be $\mathcal{N} \cup \mathcal{V}$.[16] Thus, SLCT takes *any* subtree at the bottom of the spine to be its left corner. Frozen nonterminals enable us to restrict the left corners to those labeled $\overline{\mathcal{C}}$.

**Formal guarantees.** We now discuss the formal guarantees related to our transformation in the form of an equivalence theorem (Theorem 1) and an asymptotic bound on the number of rules in the output grammar (Proposition 3). Theorem 1 establishes that the GLCT's output grammar is $\mathcal{N}$-bijectively equivalent to its input grammar.

**Theorem 1** ($\mathcal{N}$-bijective equivalence). *Suppose $\mathfrak{G} = \langle \mathcal{N}, \mathcal{V}, \mathcal{S}, \mathcal{R} \rangle$ is a WCFG and $\mathfrak{G}' = \text{GLCT}(\mathfrak{G}, \overline{\mathcal{R}}, \overline{\mathcal{C}})$ where $\overline{\mathcal{R}} \subseteq \mathcal{R}$ and $\overline{\mathcal{C}} \subseteq \mathcal{V} \cup \mathcal{N}$. Then, $\mathfrak{G} \equiv_\mathcal{N} \mathfrak{G}'$.*

We prove this theorem in Appendix C. To our knowledge, this is the only formal correctness proof for any left-corner transformation.[17] In addition, Appendix C provides pseudocode for the derivation mapping $\phi \colon \mathcal{D}_\mathcal{N} \to \mathcal{D}'_\mathcal{N}$ and its inverse $\phi^{-1}$, in Algs. 1 and 2, respectively.

We can bound the number of rules in the output grammar as a function of the input grammar and the transformation's parameters $\overline{\mathcal{C}}$ and $\overline{\mathcal{R}}$.

**Proposition 3.** *The number of rules in $\mathfrak{G}' = \text{GLCT}(\mathfrak{G}, \overline{\mathcal{R}}, \overline{\mathcal{C}})$ is no more than*

$$|\mathcal{R}| + |\mathcal{N}|\,(1 + |\overline{\mathcal{C}}| + |\overline{\mathcal{R}}|) + |\mathcal{N} \setminus \overline{\mathcal{C}}| + |\mathcal{V}|$$

*Proof.* We bound the maximum number of rules in each rule category of Def. 3:

$$|(6a)| \le |\mathcal{N} \setminus \overline{\mathcal{C}}| \qquad |(6b)| \le |\mathcal{N}|\,|\overline{\mathcal{C}}|$$
$$|(6c)| \le |\mathcal{N}| + |\mathcal{V}| \qquad |(6d)| \le |\overline{\mathcal{R}}|\,|\mathcal{N}|$$
$$|(6e)| \le |\mathcal{R} \setminus \overline{\mathcal{R}}| \qquad |(6f)| \le |\overline{\mathcal{R}}|$$

Each of these bounds can be derived straightforwardly from Def. 3. Summing them, followed by algebraic simplification, proves Proposition 3. ∎

The bound in Proposition 3 is often loose in practice, as many of the rules created by the transformation are useless. In §4, we describe how to use GLCT to eliminate left recursion, and we investigate the growth of the transformed grammar for nine natural language grammars in §4.1.

---

[15] That is, the output grammars are equal post trimming. The only useful rules are instances of (6b), (6c), and (6d). Furthermore, the (6b) rules will be useless unless $\widetilde{\alpha}$ is a terminal. With these observations, verifying that GLCT matches Johnson's (1998) presentation of LCT is straightforward.

[16] More precisely, we are using the SLCT with top-down factoring (Johnson and Roark, 2000, §2.5).

[17] We note that Aho and Ullman (1972) prove correctness for an alternative method to remove left recursion, which is used as a first step when converting a grammar to Greibach normal form. This method, however, might lead to an exponential increase in grammar size (Moore, 2000).

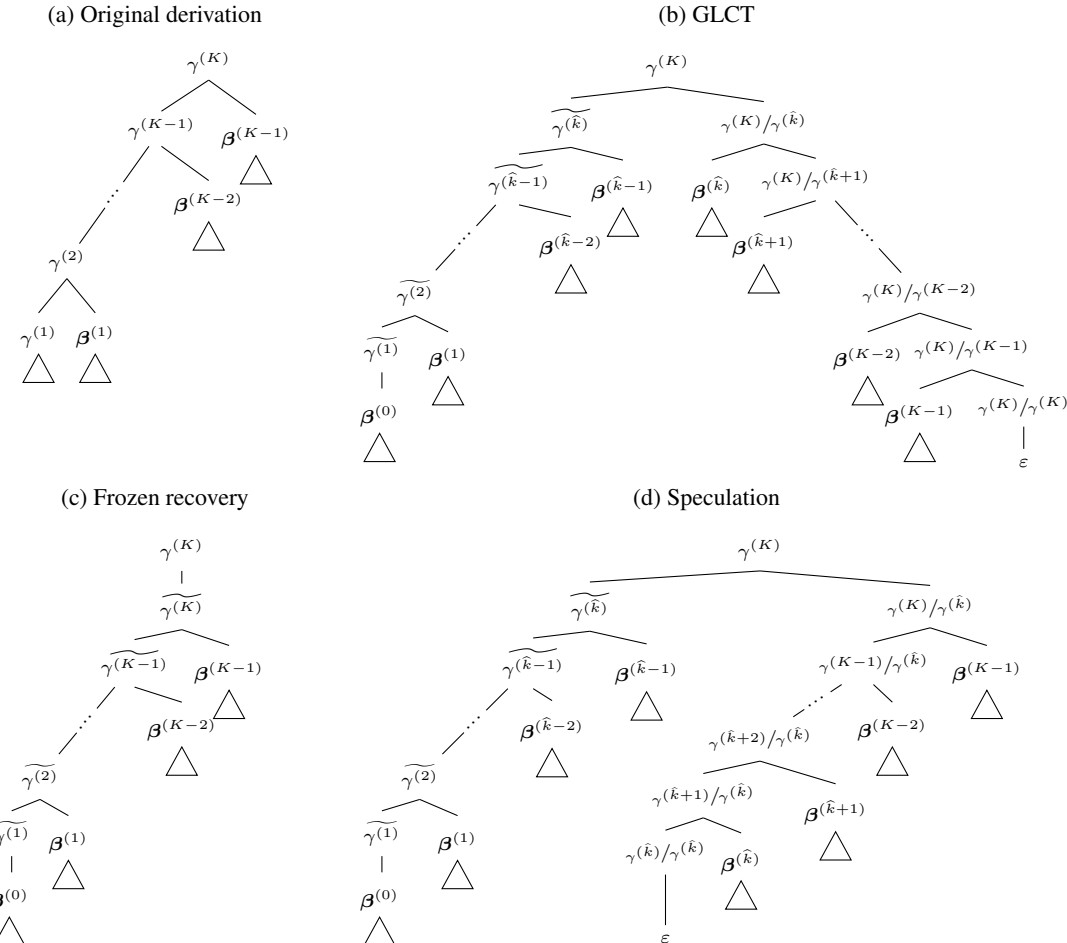

(a) Original derivation      (b) GLCT

(c) Frozen recovery      (d) Speculation

Figure 2: This figure is a schematic characterization of the one-to-one correspondence of derivations of the original grammar, speculation, and GLCT transformations. The diagram assumes that the rules exposed in the derivation $\boldsymbol{\delta}$ form its spine $\zeta(\boldsymbol{\delta})$. However, (b) and (d) assume that $\gamma^{(\widehat{k})} \in \overline{\mathcal{C}}$ is the left corner $\lambda(\boldsymbol{\delta})$, and (c) assumes that $\lambda(\boldsymbol{\delta}) = \bot$. Note that the rules in the spine that are below the left corner are frozen. When the left corner is $\bot$ (i.e., case (c)), the spines of the GLCT and speculation trees are transformed in the same manner. We note that in each transformation case ((b), (c), or (d)), each $\beta$-labeled subtree is recursively transformed, and its root label is preserved. We see that both speculation (d) and GLCT (b) hoist the same left-corner subtree (i.e., the $\gamma^{(\widehat{k})}$ subtree in the diagram) to attach at the top of the new derivation. However, the left recursion is transformed into a right-recursive tree in GLCT. Lastly, we observe that the slashed nonterminals in GLCT have a common numerator ($\gamma^{(K)}$), whereas, in speculation, they have a common denominator ($\gamma^{(\widehat{k})}$).

**Optimizations.** We briefly describe two improvements to our method's practical efficiency.

↪ **Reducing the number of useless rules.** For efficiency, we may adapt two filtering strategies from prior work that aim to reduce the number of useless rules created by the transformation.[18] We provide equations for how to modify $\mathcal{R}'$ in GLCT to account for these filters in Appendix G.

↪ **Fast nullary rule elimination.** Nullary rule elimination is often required as a preprocessing step in parsing applications (Aho and Ullman, 1972). When eliminating the nullary rules introduced by

our transformations (i.e., the base case for slashed rules), there turns out to be a special linear structure that can be exploited for efficiency. We describe the details of this speedup in Appendix H.

## 3.2 Speculation Transformation

In this section, we adapt Eisner and Blatz's (2007; §6.5) speculation transformation from weighted logic programming to WCFGs.[19] We will provide

---

[18]These strategies are provided in our implementation.

[19]The translation was direct and required essentially no invention on our behalf. However, we have made one aesthetic change to their transformations that we wish to highlight: the closest WCFG interpretation of Eisner and Blatz's (2007) speculation transformation restricts the slashed nonterminals beyond (9c) and (9d); their version constrains the denomi-

a new interpretation for speculation that does not appear in the literature.[20] In particular, we observe that speculation, like the left-corner transformation, is a subtree hoisting transformation.

**Definition 4.** *The **speculation transformation** $\textsc{spec}(\mathfrak{G},\overline{\mathcal{R}},\overline{\mathcal{C}})$ takes as input*
- *a grammar $\mathfrak{G} = \langle \mathcal{N}, \mathcal{V}, \mathcal{S}, \mathcal{R} \rangle$*
- *a subset of (non-nullary) rules $\overline{\mathcal{R}} \subseteq \mathcal{R}$ (called the left-corner recognition rules)*
- *a subset of symbols $\overline{\mathcal{C}} \subseteq (\mathcal{V} \cup \mathcal{N})$ (called the left-corner recognition symbols)*

*and outputs a grammar $\mathfrak{G}' = \langle \mathcal{N}', \mathcal{V}', \mathcal{S}', \mathcal{R}' \rangle$ with*
- *a superset of nonterminals $(\mathcal{N}' \supseteq \mathcal{N})$[21]*
- *the same set of terminals $(\mathcal{V}' = \mathcal{V})$*
- *the same start symbol $(\mathcal{S}' = \mathcal{S})$*
- *the weighted production rules $(\mathcal{R}')$:*

$$\mathrm{X} \xrightarrow{\mathbf{1}} \widetilde{\mathrm{X}}: \qquad\qquad \mathrm{X} \in \mathcal{N} \setminus \overline{\mathcal{C}} \quad (9a)$$

$$\mathrm{X} \xrightarrow{\mathbf{1}} \widetilde{\alpha}\, \mathrm{X}/\alpha: \qquad\qquad \mathrm{X} \in \mathcal{N}, \alpha \in \overline{\mathcal{C}} \quad (9b)$$

$$\mathrm{X}/\mathrm{X} \xrightarrow{\mathbf{1}} \varepsilon: \qquad\qquad \mathrm{X} \in \mathcal{V} \cup \mathcal{N} \quad (9c)$$

$$\mathrm{X}/\mathrm{Y} \xrightarrow{w} \alpha/\mathrm{Y}\,\boldsymbol{\beta}: \mathrm{X} \xrightarrow{w} \alpha\,\boldsymbol{\beta} \in \overline{\mathcal{R}}, \mathrm{Y} \in \mathcal{V} \cup \mathcal{N} \quad (9d)$$

$$\widetilde{\mathrm{X}} \xrightarrow{w} \boldsymbol{\alpha}: \qquad\qquad \mathrm{X} \xrightarrow{w} \boldsymbol{\alpha} \in \mathcal{R} \setminus \overline{\mathcal{R}} \quad (9e)$$

$$\widetilde{\mathrm{X}} \xrightarrow{w} \widetilde{\alpha}\,\boldsymbol{\beta}: \qquad \mathrm{X} \xrightarrow{w} \alpha\,\boldsymbol{\beta} \in \overline{\mathcal{R}}, \alpha \notin \overline{\mathcal{C}} \quad (9f)$$

Upon inspection, we see that the only difference between speculation and GLCT is how they define their slashed nonterminals, as the other rules are identical. The slashed nonterminals have the same base case (6c) and (9c). However, their recursive cases (6d) and (9d) differ in an intriguing way:

$$\mathrm{Y}/\alpha \xrightarrow{w} \boldsymbol{\beta}\, \mathrm{Y}/\mathrm{X}: \qquad \mathrm{X} \xrightarrow{w} \alpha\,\boldsymbol{\beta} \in \overline{\mathcal{R}}, \mathrm{Y} \in \mathcal{N} \quad (6d)$$

$$\mathrm{X}/\mathrm{Y} \xrightarrow{w} \alpha/\mathrm{Y}\,\boldsymbol{\beta}: \ \mathrm{X} \xrightarrow{w} \alpha\,\boldsymbol{\beta} \in \overline{\mathcal{R}}, \mathrm{Y} \in \mathcal{V} \cup \mathcal{N} \quad (9d)$$

This difference is why GLCT can eliminate left recursion and speculation cannot: GLCT's slashed nonterminal appears to the right of $\boldsymbol{\beta}$, and speculation's appears on the left. For GLCT, Y is passed along the numerator of the slashed nonterminal,

whereas, for speculation, Y is passed along the denominator. Theorem 2 (below) establishes that speculation and GLCT are bijectively equivalent for their complete set of nonterminals.[22]

**Theorem 2** (Speculation–GLCT bijective equivalence)**.** *For any grammar $\mathfrak{G}$, and choice of $\overline{\mathcal{C}}$ and $\overline{\mathcal{R}}$, $\textsc{spec}(\mathfrak{G},\overline{\mathcal{R}},\overline{\mathcal{C}})$ and $\textsc{glct}(\mathfrak{G},\overline{\mathcal{R}},\overline{\mathcal{C}})$ are $\mathcal{X}$-bijectively equivalent where $\mathcal{X}$ is the complete set of symbols (i.e., original, frozen, and slashed).*

See Appendix D for the proof sketch. We also provide the first proof of equivalence for speculation.

**Theorem 3.** *For any grammar $\mathfrak{G} = \langle \mathcal{N}, \mathcal{V}, \mathcal{S}, \mathcal{R} \rangle$, $\overline{\mathcal{R}} \subseteq \mathcal{R}$, and $\overline{\mathcal{C}} \subseteq \mathcal{V} \cup \mathcal{N}$: $\textsc{spec}(\mathfrak{G},\overline{\mathcal{R}},\overline{\mathcal{C}}) \equiv_{\mathcal{N}} \mathfrak{G}$.*

*Proof.* The theorem follows directly from Theorem 1, Theorem 2, and the compositionality of bijective functions. ∎

## 4 Left-Recursion Elimination

Motivated by the desire for efficient top-down parsing for which left-recursion poses challenges (§1), we describe how GLCT may be used to transform a possibly left-recursive grammar $\mathfrak{G}$ into a bijectively equivalent grammar $\mathfrak{G}'$ without left-recursion.[23] The bijective equivalence (Theorem 1) ensures that we can apply an inverse transformation to the derivation tree of the transformed grammar into its corresponding derivation tree in the original grammar. This section provides an efficient and (provably correct) recipe for left-recursion elimination using GLCT. We experiment with this recipe on natural language grammars in §4.1.

Our left-recursion elimination recipe is based on a single application of GLCT, which appropriately chooses parameters $\overline{\mathcal{C}}$ and $\overline{\mathcal{R}}$. We describe how to determine these parameters by analyzing the structure of the rules in $\mathfrak{G}$.

We define the **left-recursion depth** of a derivation tree $d(\boldsymbol{\delta})$ as the length of the path from the root to the leftmost leaf node. The left-recursion depth of a grammar is $d(\mathfrak{G}) \overset{\text{def}}{=} \max_{\boldsymbol{\delta} \in \mathcal{D}} d(\boldsymbol{\delta})$. We say that $\mathfrak{G}$ is **left-recursive** iff $d(\mathfrak{G})$ is unbounded. To analyze whether $\mathfrak{G}$ is left-recursive, we can analyze its left-recursion graph, which accurately characterizes the left-recursive paths from the root of a derivation to its leftmost leaf in the set of derivations $\mathcal{D}$. The **left-recursion graph** of the grammar

---

nator to be $\in \overline{\mathcal{C}}$. This difference disappears after trimming because *useful* slashed nonterminals must be consumed by the recovery rule (9b), which imposes the $\overline{\mathcal{C}}$ constraint one level higher. We prefer our version as it enables Theorem 2, which shows that GLCT and speculation produce $\mathcal{X}$-bijective equivalent grammars for all nonterminals. The pruned version would result in a weaker theorem with $\mathcal{X}$ being a *subset* of the nonterminals with a nuanced specification.

[20]Vieira's (2023) dissertation, which appeared contemporaneously with this paper, adopts our same interpretation.

[21]See footnote 12.

[22]We note that the set of *useful* slashed and frozen nonterminals typically differs between GLCT and speculation.

[23]Note: when we say left recursion, we often mean *unbounded* left recursion.

$\mathfrak{G}$ is a labeled directed graph $G = \langle N, E \rangle$ with nodes $N = \mathcal{V} \cup \mathcal{N}$ and edges $E = \{(X \xrightarrow{r} \alpha) \mid r \in \mathcal{R}, r = (X \rightarrow \alpha \cdots)\}$. It should be clear that the $\mathfrak{G}$ is left-recursive iff $G$ has a cyclic subgraph. We classify a rule $r$ as left-recursive if the edge labeled $r$ is an edge in any cyclic subgraph of $G$. To determine the set of left-recursive rules, we identify the strongly connected components (SCCs) of $G$ (e.g., using Tarjan's (1972) algorithm). The SCC analysis returns a function $\pi$ that maps each of $G$'s nodes to the identity of its SCC. Then, a rule $r$ is left-recursive iff its corresponding edge $\alpha \xrightarrow{r} \beta$ satisfies $\pi(\alpha) = \pi(\beta)$.[24] To ensure that left recursion is eliminated, $\overline{\mathcal{R}}$ must include all left-recursive rules.

We use the following set to provide a sufficient condition on $\overline{\mathcal{C}}$ to eliminate left recursion:

$$\text{bottoms}(\overline{\mathcal{R}}) \overset{\text{def}}{=} \qquad (10)$$
$$(\mathcal{V} \cup \{X \mid (X \xrightarrow{r} \alpha) \in E, r \in \mathcal{R} \setminus \overline{\mathcal{R}}\})$$
$$\cap \{\alpha \mid (X \rightarrow \alpha \cdots) \in \overline{\mathcal{R}}\})$$

This set captures the set of nodes that may appear at the bottom of a spine (for the given $\overline{\mathcal{R}}$). This is because the spine is defined as the longest sequence of rules in $\overline{\mathcal{R}}$ along the left of a derivation; thus, a spine can end in one of two ways (1) it reaches a terminal, or (2) it encounters a rule outside of $\overline{\mathcal{R}}$. Thus, the bottom elements of the spine are the set of terminals, and the set of nodes with at least one $(\mathcal{R} \setminus \overline{\mathcal{R}})$-labeled outgoing edge—which we refine to nodes that might appear in the spine (i.e., those in the leftmost position of the rules in $\overline{\mathcal{R}}$).

With these definitions in place, we can provide sufficient conditions on the GLCT parameter sets that will remove left recursion:

**Theorem 4** (Left-recursion elimination). *Suppose that $\mathfrak{G}' = \text{TRIM}(\text{GLCT}(\mathfrak{G}, \overline{\mathcal{R}}, \overline{\mathcal{C}}))$ where*
- *$\mathfrak{G}$ has no unary rules*
- *$\overline{\mathcal{R}} \supseteq$ the left-recursive rules in $\mathfrak{G}$*
- *$\overline{\mathcal{C}} \supseteq \text{bottoms}(\overline{\mathcal{R}})$*

*Then, $\mathfrak{G}'$ is not left-recursive. Moreover, $d(\mathfrak{G}') \leq 2 \cdot C$ where $C$ is the number of SCCs in the left-recursion graph for $\mathfrak{G}$.*

See Appendix E for the proof.

**Example.** In Fig. 1, we made use of Theorem 4 to remove the left-recursion from NP to NP, applying GLCT with $\overline{\mathcal{C}} = \{\text{NP}\}$ and $\overline{\mathcal{R}} = \{\mathcal{S} \rightarrow \text{NP VP},$ $\text{NP} \rightarrow \text{PossP NN}, \text{PossP} \rightarrow \text{NP 's}\}$.[25]

---

[24]Note that nullary rules cannot be left-recursive.

[25]Note that omitting $\mathcal{S} \rightarrow \text{NP VP}$ from $\overline{\mathcal{R}}$ would have also eliminated left recursion in this example, but we would have

**Our recipe.** We take $\overline{\mathcal{R}}$ as the set of left-recursive rules in $\mathfrak{G}$ and $\overline{\mathcal{C}}$ as $\text{bottoms}(\overline{\mathcal{R}})$.[26] This minimizes the upper bound given by Proposition 3 subject to the constraints given in Theorem 4.

**Special cases.** Theorem 4 implies that the basic left-corner transformation and the selective left-corner transformations (with $\overline{\mathcal{R}} \supseteq$ the left-recursive rules) will eliminate left recursion. Experimentally, we found that our recipe produces a slightly smaller grammar than the selective option (see §4.1).

**Unary rules.** The reason Theorem 4 requires that $\mathfrak{G}$ is unary-free is that the left-corner transformation cannot remove unary cycles of this type.[27] To see why, note that $\beta = \varepsilon$ for a unary rule (6d); thus, the transformed rule will have a slashed nonterminal in its leftmost position, so it may be left-recursive. Fortunately, unary rule cycles can be eliminated from WCFGs by standard preprocessing methods (e.g., Stolcke (1995, §4.5) and Opedal et al. (2023, §E)). However, we note that eliminating such unary chain cycles does not produce an $\mathcal{N}$-bijectively equivalent grammar as infinitely many derivations are mapped to a single one that accounts for the total weight of all of them.

**Nullary rules.** We also note that the case where the $\mathfrak{G}$ may derive $\varepsilon$ as its leftmost constituent also poses a challenge for top-down parsers. For example, that would be the case in Fig. 1 if $\text{PossP} \rightarrow \text{NP POS}$ was replaced by $\text{PossP} \rightarrow \text{X NP POS}$ and $\text{X} \rightarrow \varepsilon$; this grammar is not left-recursive, but the subgoal of recognizing an NP in top-down parser will still, unfortunately, lead to infinite recursion. Thus, a complete solution to transforming a grammar into a top-down-parser-friendly grammar should also treat these cases. To that end, we can transform the original grammar into an equivalent nullary-free version with standard methods (Opedal et al., 2023, §F) before applying our GLCT-based left-recursion elimination recipe. As with unary rule elimination, nullary rule elimination does not produce an $\mathcal{N}$-bijectively equivalent grammar.

## 4.1 Experiments

In this section, we investigate how much the grammar size grows in practice when our GLCT recipe is used to eliminate left recursion. We compare our

---

obtained a different output tree in the figure.

[26]Our choice for $\overline{\mathcal{R}}$ is consistent with the recommendation for SLCT in Johnson and Roark (2000).

[27]Prior left-corner transformations (Johnson and Roark, 2000; Moore, 2000) are limited in the same manner.

| Language (size) | Method | Raw | +Trim | $-\varepsilon$'s |
|---|---|---|---|---|
| Basque (73,173) | SLCT | 3,354,445 | 245,989 | 411,023 |
| | GLCT | 644,125 | 245,923 | 411,023 |
| English (21,272) | SLCT | 514,338 | 26,655 | 46,088 |
| | GLCT | 203,664 | 26,289 | 46,088 |
| French (105,896) | SLCT | 5,902,552 | 106,860 | 173,375 |
| | GLCT | 147,860 | 106,628 | 173,375 |
| German (100,346) | SLCT | 21,204,060 | 272,406 | 434,787 |
| | GLCT | 1,930,386 | 271,752 | 434,787 |
| Hebrew (84,648) | SLCT | 14,304,366 | 564,456 | 979,040 |
| | GLCT | 2,910,538 | 564,074 | 979,040 |
| Hungarian (134,461) | SLCT | 17,823,603 | 151,373 | 242,360 |
| | GLCT | 748,398 | 151,013 | 242,360 |
| Korean (59,557) | SLCT | 54,575,826 | 87,937 | 96,023 |
| | GLCT | 3,706,444 | 86,529 | 96,023 |
| Polish (41,957) | SLCT | 2,845,430 | 61,333 | 79,341 |
| | GLCT | 177,610 | 61,253 | 79,341 |
| Swedish (79,137) | SLCT | 20,483,899 | 1,894,346 | 3,551,917 |
| | GLCT | 6,896,791 | 1,871,789 | 3,551,917 |

Table 1: Results of applying GLCT and SLCT on the ATIS and SPMRL grammars broken down by each of the nine languages. We present the resulting size in the raw output grammar (Raw), trimming (+Trim), and binarization plus nullary removal ($-\varepsilon$'s).

results to SLCT with top-down factoring (§3.1) to see whether the additional degree of freedom given by $\overline{\mathcal{C}}$ leads to any reduction in size. We apply both transformations to nine grammars of different languages: Basque, English, French, German, Hebrew, Hungarian, Korean, Polish, and Swedish. We use the ATIS grammar (Dahl et al., 1994) as our English grammar.[28] We derived the other grammars from the SPMRL 2013/2014 shared tasks treebanks (Seddah et al., 2013, 2014).[29]

**Experimental setup.** For GLCT, we set $\overline{\mathcal{R}}$ and $\overline{\mathcal{C}}$ according to our recipe. For SLCT, we set $\overline{\mathcal{R}}$ according to Theorem 4's conditions for removing left-recursion. We compare the grammar size and the number of rules of the raw output grammar to those of the input grammar. However, the raw output sizes can be reduced using useless rule filters (discussed in §3.1 and Appendix G),

so we additionally apply trimming to the output grammars. When parsing, it is often practical to first binarize the grammar and remove nullary rules, so we perform those postprocessing steps as well.

As a sanity check, we verify that left recursion is removed in all settings by checking that the left-recursion graph of the output grammar is acyclic. We present the results as evaluated on grammar size in Table 1. Appendix F provides further results in terms of the number of rules.

**Discussion.** Interestingly, the increase in size compared to the input grammar varies a lot between languages. Previous work (Johnson and Roark, 2000; Moore, 2000) only evaluated on English and thus appear to have underestimated the blow-up caused by the left-corner transformation when applied to natural language grammars. Compare, for instance, the ratio between the trimmed size and the original size in Table 1 of English (1.2) to Basque (3.4), Hebrew (6.7), and Swedish (23.7). By Proposition 3, the number of rules in the output grammar scales with $|\overline{\mathcal{R}}|$, which by Theorem 4 is set as the left-recursive rules.

The GLCT produces smaller grammars than the SLCT for all languages before either of the postprocessing steps. This difference is (almost) eliminated post-trimming, however, which is unsurprising given that SLCT is a special case of GLCT (§3.1). The small difference in size after trimming happens since two rules of the form $X \to \widetilde{X}\ X/X$ (6b) and $X/X \to \varepsilon$ (6c) in SLCT are replaced by one rule $X \to \widetilde{X}$ (6a) in GLCT. However, this difference disappears after nullary removal.

## 5 Conclusion

This work generalized the left-corner transformation to operate not only on a subset of rules but also on a subset of nonterminals. We achieve this by adapting frozen nonterminals from the speculation transformation. We exposed a tight connection between generalized left-corner transformation and speculation (Theorem 2). Finally, and importantly, we proved the transformation's correctness (Theorem 1) and provided precise sufficient conditions for when it eliminates left recursion (Theorem 4).

## Limitations

Parsing runs in time proportional to the grammar size. §4.1 shows that we obtain the same grammar size after postprocessing from our method and the

---

[28]We selected the (boolean-weighted) ATIS grammar because it was used in prior work (Moore, 2000). We note, however, that—despite our best efforts—we were unable to replicate Moore's (2000) *exact* grammar size on it.

[29]Specifically, we load all trees from the SPMRL 5k training dataset, delete the morphological annotations, collapse unary chains like $X \to Y \to Z \to \alpha$ into $X \to \alpha$, and create a grammar from the remaining rules. The weights of the SPMRL grammars are set using maximum-likelihood estimation. None of the treebanks contained nullary rules.

selective left-corner transformation, which gave us no reason to provide an empirical comparison on parsing runtime.

Moreover, it is thus of practical importance to restrict the growth of the grammar constant. We have discussed a theoretical bound for grammar growth in Proposition 3, investigated it empirically in §4.1, and provided further tricks to reduce it in Appendix G. Orthogonally to the left-corner transformation itself, it is possible to factor the grammar so that the grammar size is minimally affected by the transformation. Intuitively, reducing the number of left-recursive rules in $\mathcal{R}$ will also reduce the number of rules that are required in $\overline{\mathcal{R}}$, which, in turn, leads to fewer rules in the output grammar. We did not present any such preprocessing techniques here, but Moore (2000) provides a reference for two methods: left-factoring and non-left-recursive grouping. Johnson and Roark (2000) give a second factoring trick in addition to top-down factoring (see footnote 16), which is similar to Moore's (2000) left-factoring. We also mention that Vieira's (2023) search-based technique for optimizing weighted logic programs could be directly applied to grammars. In particular, the search over sequences of define–unfold–fold transformations can be used to find a smaller grammar that encodes the same weighted language.

There appear to be connections between our notion of slashed nonterminals and the left quotient of formal languages that we did not explore in this paper.[30] For example, the simplest case of the left quotient is the Brzozowski (1964) derivative. The Brzozowski derivative of $\mathfrak{G}$ with respect to $a \in \mathcal{V}$ is equal to the weighted language of $\mathcal{S}/a$ in the output grammar $\mathfrak{G}'$ produced by speculation or GLCT, provided that the $\mathfrak{G}$ is nullary-free, $\overline{\mathcal{R}} = \mathcal{R}$, and $a \in \overline{\mathcal{C}}$. We suspect that other interesting connections are worth formalizing and exploring further.

Finally, we note that we could extend our transformation to deal with nullary rules directly rather than eliminating them by preprocessing (as discussed in §4). The idea is to modify (6d) in GLCT so that the slashed nonterminal on its right-hand side is formed from the leftmost nonterminal that derives something *other than* $\varepsilon$, rather than the leftmost symbol. For this extension to work out, we require that the grammar is preprocessed such that

each nonterminal is replaced by two versions: one that generates only $\varepsilon$, and one that generates anything else. Preprocessing the grammar in this way is also done in nullary rule elimination (see Opedal et al. (2023), §F) for details.

## Ethical Statement

We do not foresee any ethical issues with our work.

## Acknowledgments

We thank Alex Warstadt, Clemente Pasti, Ethan Wilcox, Jason Eisner, and Josef Valvoda for valuable feedback and discussions. We especially thank Clemente for pointing out the nullary removal optimization in Proposition 4. We also thank the reviewers for their useful comments, suggestions, and references to related work. Andreas Opedal acknowledges funding from the Max Planck ETH Center for Learning Systems.

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

## A  Proof of Proposition 1 (Decomposition)

**Proposition 1** (Decomposition). *Suppose $\mathfrak{G}' = \text{GLCT}(\mathfrak{G}, \overline{\mathcal{R}}, \overline{\mathcal{C}})$. Then, for any $X \in \mathcal{N}$:*

$$\mathfrak{G}_X = \mathfrak{G}'_{\widetilde{X}} \oplus \bigoplus_{\alpha \in \overline{\mathcal{C}}} \mathfrak{G}'_{\widetilde{\alpha}} \circ \mathfrak{G}'_{X/\alpha} \tag{7}$$

*Proof.*

$$\mathfrak{G}_X = \mathfrak{G}'_X \qquad\qquad\qquad [\text{by Theorem 1}] \tag{11}$$

$$= \bigoplus_{(X \xrightarrow{w} \beta_1 \cdots \beta_K) \in \mathcal{R}'} w \circ \mathfrak{G}'_{\beta_1} \circ \cdots \circ \mathfrak{G}'_{\beta_K} \qquad\qquad [\text{by Eq. 5}] \tag{12}$$

$$= \bigoplus_{(X \xrightarrow{1} \widetilde{X}) \in \mathcal{R}'} \mathbf{1} \circ \mathfrak{G}'_{\widetilde{X}} \oplus \bigoplus_{(X \xrightarrow{1} \widetilde{\alpha}\, X/\alpha) \in \mathcal{R}'} \mathbf{1} \circ \mathfrak{G}'_{\widetilde{\alpha}} \circ \mathfrak{G}'_{X/\alpha} \qquad [\text{by Def. 3}] \tag{13}$$

$$= \mathfrak{G}'_{\widetilde{X}} \oplus \bigoplus_{\alpha \in \overline{\mathcal{C}}} \mathfrak{G}'_{\widetilde{\alpha}} \circ \mathfrak{G}'_{X/\alpha} \qquad\qquad [\text{by Def. 3 and algebra}] \tag{14}$$

Note that (13) specializes the sum to the only kinds of rules that can build $X \in \mathcal{N}$: rules (6a) and (6b).  ∎

## B  Proof of Proposition 2 (Weighted Language Relationship)

**Lemma 1.** $\forall \alpha \in \mathcal{N} \cup \mathcal{V}$, *there exists a weight- and yield- preserving bijection between* $\mathcal{D}_\alpha / \perp$ *and* $\mathcal{D}'_{\widetilde{\alpha}}$.

*Proof (Sketch).* On the left, we have a prototypical derivation from $\mathcal{D}_\alpha / \perp$ with its spine exposed. Here $\alpha = \gamma^{(K)}$. By definition, the spine elements $\gamma^{(1)}, \dots \gamma^{(K)} \notin \overline{\mathcal{C}}$. On the right, we have its corresponding derivation in $\mathcal{D}'_{\widetilde{\alpha}}$:

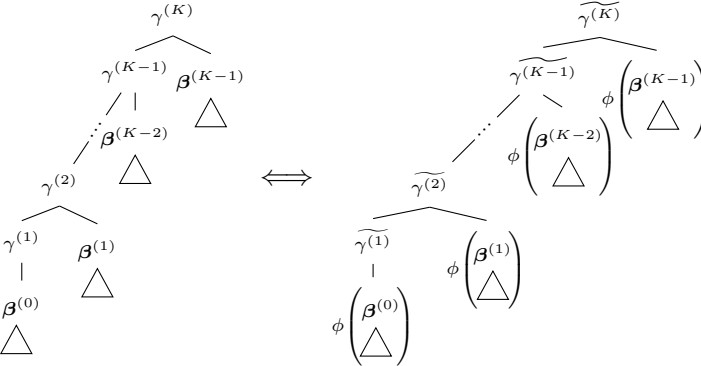

where the function $\phi$ is Alg. 1. Call the function that maps the left tree to the right one $\psi$; all it does is freeze the spine and then call $\phi$ on the $\boldsymbol{\beta}$ subtrees. Recall that Theorem 1 established that $\phi$ is a $\mathcal{N}$-structure-preserving bijection. Thus, it is straightforward to see that $\psi$ is also weight- and yield-preserving bijection, as its inverse $\psi^{-1}$ undoes these steps and calls $\phi^{-1}$ on the subtrees. This proves Lemma 1. ∎

**Lemma 2.** $\forall X \in \mathcal{N}, \alpha \in \mathcal{N} \cup \mathcal{V}$, *there exists a weight- and yield-preserving bijection between* $\mathcal{D}_X / \alpha$ *and* $\mathcal{D}'_{X/\alpha}$.

*Proof (Sketch).* On the left, we have a prototypical derivation from $\mathcal{D}_X / \alpha$ with its spine exposed. Recall that these trees result from replacing the left corner, which is why there is an $\varepsilon$. Here $X = \gamma^{(K)}$ and $\alpha = \gamma^{(\widehat{k})}$. On the right, we have its corresponding derivation in $\mathcal{D}'_{X/\alpha}$.

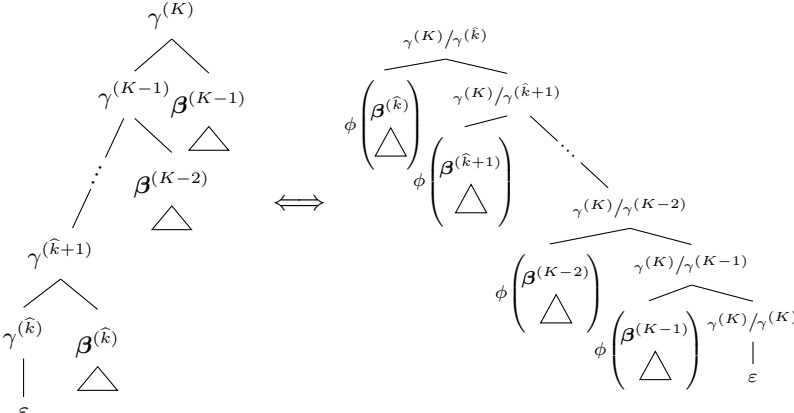

where the function $\phi$ is Alg. 1. Call the mapping from left to right $\psi$; it is very straightforward. All it does is (1) transpose, (2) relabel the spine, and (3) call $\phi$ on the $\boldsymbol{\beta}$ subtrees. Note that $\phi$ does *not* recurse to $\psi$ because it can already convert the $\boldsymbol{\beta}$ subtrees by Theorem 1. Thus, it is straightforward to see that $\psi$ is also weight- and yield-preserving bijection. This proves Lemma 2. ∎

**Proposition 2** (Weighted language relationship). *Suppose* $\mathfrak{G}' = \text{GLCT}(\mathfrak{G}, \overline{\mathcal{C}}, \overline{\mathcal{R}})$. *Then, for any* $X \in \mathcal{N}$ *and* $\alpha \in \mathcal{N} \cup \mathcal{V}$:

$$\mathfrak{G}'_{\widetilde{\alpha}} = \bigoplus_{\boldsymbol{\delta} \in \mathcal{D}_\alpha / \perp} [\![\boldsymbol{\delta}]\!] \quad and \quad \mathfrak{G}'_{X/\alpha} = \bigoplus_{\boldsymbol{\delta} \in \mathcal{D}_X / \alpha} [\![\boldsymbol{\delta}]\!]$$

*Proof.* The equations follow directly from Lemma 1 and Lemma 2. In each case, the left- and the right-hand sides are sums of the derivations that are part of a weight- and yield-preserving bijection. ∎

## C  Proof of Theorem 1 (Bijective Equivalence)

**Roadmap.**  We will show that $\mathfrak{G} \equiv_{\mathcal{N}} \text{GLCT}(\mathfrak{G}, \overline{\mathcal{R}}, \overline{\mathcal{C}})$ for any choice of $\overline{\mathcal{C}}$ and $\overline{\mathcal{R}}$. Our proof makes use of two lemmas, Lemma 3 and Lemma 4, to establish that the derivation mapping $\phi$ (Alg. 1) and its inverse $\phi^{-1}$ (Alg. 2) define a bijection of the necessary type. Lemma 3 shows that $\phi$ preserves label, weight, and yield. Lemma 4 shows that $\phi$ is invertible and, thus, that a bijection exists.

**Algorithm 1** Derivation mapping for the generalized left-corner transformation $\text{GLCT}(\mathfrak{G}, \overline{\mathcal{R}}, \overline{\mathcal{C}})$

1.  **def** $\phi(\boldsymbol{\delta})$:
2.      **if** $\boldsymbol{\delta} \in \mathcal{V}$ :          $\triangleright$*Base case*
3.          **return** $\boldsymbol{\delta}$
4.      **else if** $\boldsymbol{\delta}$ is a sequence $\boldsymbol{\delta}_1 \cdots \boldsymbol{\delta}_K$ :          $\triangleright$*Sequence case*
5.          **return** $\phi(\boldsymbol{\delta}_1) \cdots \phi(\boldsymbol{\delta}_K)$
6.      **else**
7.  

        $\leftarrow \boldsymbol{\delta}$   **where** $(\gamma^{(K)} \to \gamma^{(K-1)} \boldsymbol{\beta}^{(K-1)}) \cdots (\gamma^{(2)} \to \gamma^{(1)} \boldsymbol{\beta}^{(1)})$ is $\zeta(\boldsymbol{\delta})$ the spine of the $\boldsymbol{\delta}$

8.      $\widehat{k} \leftarrow \min\big(\{i \mid \gamma^{(i)} \in \overline{\mathcal{C}}, i \in \{1, \dots K\}\}\big).$      $\triangleright$*determine the left corner; take* $\min(\emptyset) = \infty$
9.      **if** $\widehat{k} = \infty$ :
10.         **return**

11.     **else**
12.         **return**

**Algorithm 2** Inverse derivation mapping for the generalized left-corner transformation $\textsc{glct}(\mathfrak{G}, \overline{\mathcal{R}}, \overline{\mathcal{C}})$

1. **def** $\phi^{-1}(\delta')$:
2.   **if** $\delta' \in \mathcal{V}$ :                                                                                        ▷*Base case*
3.     **return** $\delta'$
4.   **else if** $\delta'$ is a sequence $\delta'_1 \cdots \delta'_K$ :                                                        ▷*Sequence case*
5.     **return** $\phi^{-1}(\delta'_1) \cdots \phi^{-1}(\delta'_K)$
6.   **else if** top rule of $\delta'$ is an instance of (6b) :        ▷*Slashed recovery rule; $\delta'$ must have the following form:*

7. 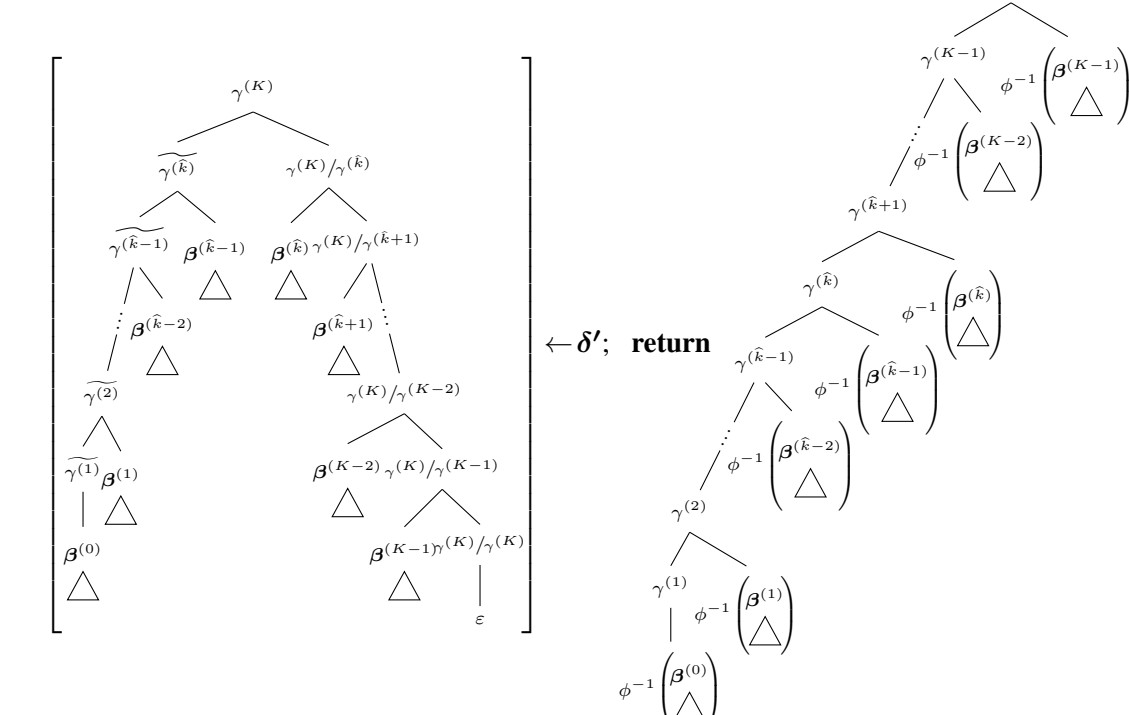

8.   **else**                                   ▷*Frozen recovery rule; top rule of $\delta'$ is an instance (6a) and $\delta'$ must have the following form:*

9. 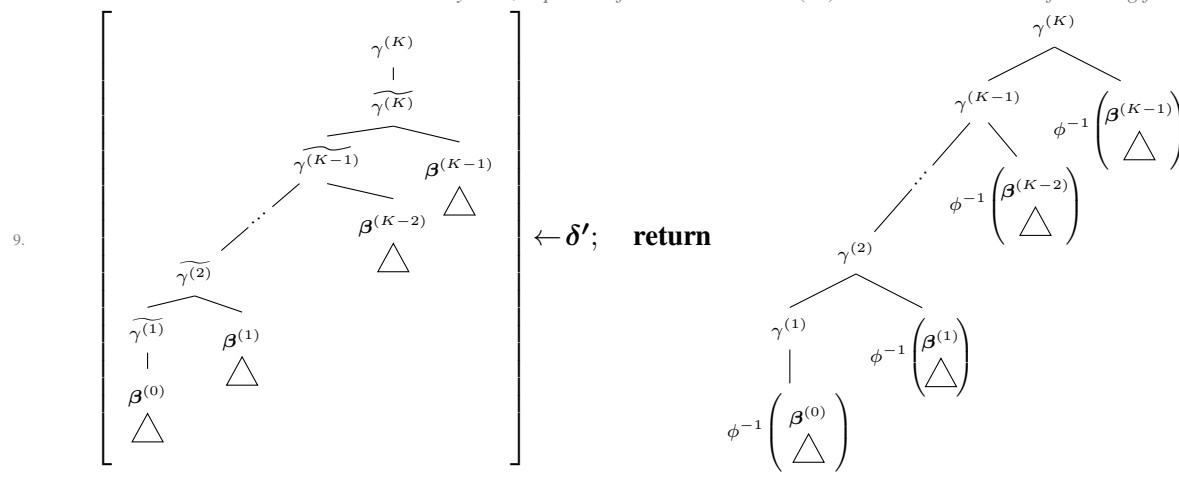

**Lemma 3.** *Let $\mathfrak{G} = \langle \mathcal{N}, \mathcal{V}, \mathcal{S}, \mathcal{R}, w \rangle$ be a WCFG. Let $\mathfrak{G}' = \textsc{glct}(\mathfrak{G}, \overline{\mathcal{R}}, \overline{\mathcal{C}})$ where $\overline{\mathcal{C}} \subseteq (\mathcal{N} \cup \mathcal{V})$ and $\overline{\mathcal{R}} \subseteq \mathcal{R}$. Then, Alg. 1 defines a function $\phi$ that is*

*(i) type $\phi \colon \mathcal{D} \to \mathcal{D}'$*

*(ii) label-preserving*

*(iii) yield-preserving*

*(iv) weight-preserving*

*Proof.* Let $\mathfrak{G}' = \langle \mathcal{N}', \mathcal{V}', \mathcal{S}', \mathcal{R}' \rangle$. Note: by construction (Def. 3), $\mathcal{N}' \supseteq \mathcal{N}$, $\mathcal{S}' = \mathcal{S}$, and $\mathcal{V}' = \mathcal{V}$.

For notational convenience, we extend $\omega$ and $\sigma$ to apply to a sequence of derivations $\underset{\triangle}{\alpha_1} \cdots \underset{\triangle}{\alpha_K} \in \mathcal{D}^*$:

$$\omega\left(\underset{\triangle}{\alpha_1} \cdots \underset{\triangle}{\alpha_K}\right) \overset{\text{def}}{=} \omega\left(\underset{\triangle}{\alpha_1}\right) \otimes \cdots \otimes \omega\left(\underset{\triangle}{\alpha_K}\right) \quad \text{and} \quad \sigma\left(\underset{\triangle}{\alpha_1} \cdots \underset{\triangle}{\alpha_K}\right) \overset{\text{def}}{=} \sigma\left(\underset{\triangle}{\alpha_1}\right) \circ \cdots \circ \sigma\left(\underset{\triangle}{\alpha_K}\right)$$

It is straightforward to verify that $\phi$ satisfies properties (i–iv) for sequences of derivations; for brevity, we will not do so.

Fix an arbitrary derivation tree $\boldsymbol{\delta} \in \mathcal{D}$. Our proof will proceed by structural induction on the **subtree relation**: $\boldsymbol{\delta}_1 \prec \boldsymbol{\delta} \iff \boldsymbol{\delta}_1$ is a (strict) subtree of $\boldsymbol{\delta}$. It is well-known that $\prec$ is well-founded ordering, making it suitable for induction (e.g., Burstall, 1969).

**Inductive hypothesis (IH)**: $\forall \boldsymbol{\delta}_1 \prec \boldsymbol{\delta}$: properties (i–iv) hold.

**Base case ($\boldsymbol{\delta} \in \mathcal{V}$)**: Here $\phi$ is the identity mapping. Thus, properties (i–iv) are clearly preserved.

**Inductive case ($\boldsymbol{\delta} \notin \mathcal{V}$)**: Here $\boldsymbol{\delta}$ must have the general form:

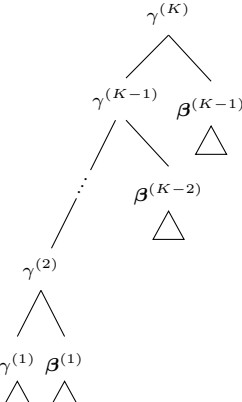

where $K \geq 1$, $\forall i \in \{1, \dots, K\} \colon \boldsymbol{\beta}^{(i)} \in (\mathcal{V} \cup \mathcal{N})^*$, and $(\gamma^{(K)} \to \gamma^{(K-1)} \boldsymbol{\beta}^{(K-1)}) \cdots (\gamma^{(2)} \to \gamma^{(1)} \boldsymbol{\beta}^{(1)})$ is $\zeta(\boldsymbol{\delta})$, the spine of the $\boldsymbol{\delta}$.

Observation $\boxed{1}$: The subtree $\underset{\triangle}{\gamma^{(1)}}$ is either a terminal or it is of the form $\begin{matrix} \gamma^{(1)} \\ | \\ \underset{\triangle}{\boldsymbol{\beta}^{(0)}} \end{matrix}$ with $\gamma^{(1)} \to \boldsymbol{\beta}^{(0)} \notin \overline{\mathcal{R}}$.

Let $\widehat{k}$ be the index of the left corner (i.e., bottom-most element of $\overline{\mathcal{C}}$ that occurs along the spine of $\boldsymbol{\delta}$).[31] More formally, let $\widehat{k} = \min(\{i \mid \gamma^{(i)} \in \overline{\mathcal{C}}, i \in \{1, \dots K\}\})$. If no such element exists, define $\widehat{k} = \min(\emptyset) = \infty$. The cases correspond to $\lambda(\boldsymbol{\delta}) \neq \bot$ and $\lambda(\boldsymbol{\delta}) = \bot$, respectively.

We consider two cases: $\widehat{k} = \infty$ and $\widehat{k} < \infty$:

---

[31]We chose the notation $\widehat{k}$ because the $\widehat{\phantom{\cdot}}$ is visually similar to the "kink" in the transformed tree that is determined by $\widehat{k}$ in the case where $\widehat{k} < \infty$.

**Case** ($\widehat{k} = \infty$):    In this case, we observe $\boxed{2}$: $\forall i \in \{1, \dots, K\} : \gamma^{(i)} \notin \overline{\mathcal{C}}$, and line 10 of Alg. 1 returns:

$$\phi(\boldsymbol{\delta}) =$$

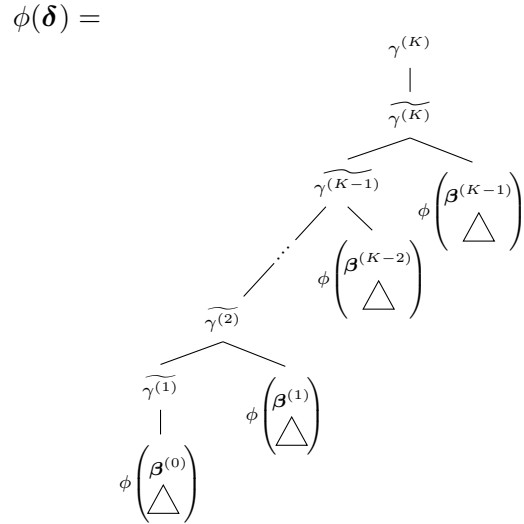

We check each of the properties:

(i) $\phi \colon \mathcal{D} \to \mathcal{D}'$: In this case, we can check that $\phi(\boldsymbol{\delta}) \in \mathcal{D}'$ by verifying that each rule used in the definition of $\phi$ appears in $\mathfrak{G}'$, and then invoking the inductive hypothesis on the subtrees.

It is straightforward to verify that the mapping $\phi(\boldsymbol{\delta})$ preserves weight and that the tree exists in $\mathcal{D}'$ by crosschecking each of the rules that appear in it with the GLCT construction (Def. 3):

- The top rule $\gamma^{(K)} \to \widetilde{\gamma^{(K)}}$ is an instance of (6a). The side condition $X \in \mathcal{N} \setminus \overline{\mathcal{C}}$ is satisfied by $\boxed{2}$ ($\forall i \in \{1, \dots, K\} : \gamma^{(i)} \notin \overline{\mathcal{C}}$).

- Each middle rule $\widetilde{\gamma^{(i)}} \to \widetilde{\gamma^{(i-1)}} \boldsymbol{\beta}^{(i-1)}$ is an instance of (6f). The side condition $X \xrightarrow{w} X_1 \boldsymbol{\beta} \in \overline{\mathcal{R}}, X_1 \notin \overline{\mathcal{C}}$ is satisfied in each case because spine guarantees inclusion in $\overline{\mathcal{R}}$, and by $\boxed{2}$ ($\forall i \in \{1, \dots, K\} : \gamma^{(i)} \notin \overline{\mathcal{C}}$).

- The bottom rule $\widetilde{\gamma^{(1)}} \to \boldsymbol{\beta}^{(0)}$ is an instance of (6e). The side condition $X \xrightarrow{w} \boldsymbol{\beta} \notin \overline{\mathcal{R}}$ is satisfied by $\boxed{1}$.[32]

Since we have verified that each of the rules in the construction of $\phi(\boldsymbol{\delta})$ are in $\mathfrak{G}'$, and the subtrees are in $\mathcal{D}'$ by IH, then it follows that $\phi(\boldsymbol{\delta}) \in \mathcal{D}'$.

(ii) label-preserving: It is clear by inspection that $\boldsymbol{\delta}$ and $\phi(\boldsymbol{\delta})$ have the same root label.

(iii) yield-preserving:

$$\sigma(\boldsymbol{\delta}) = \sigma\left( \begin{array}{c} \boldsymbol{\beta}^{(0)} \\ \triangle \end{array} \right) \circ \cdots \circ \sigma\left( \begin{array}{c} \boldsymbol{\beta}^{(K-1)} \\ \triangle \end{array} \right) \qquad \text{[definition]}$$

$$= \sigma\left( \phi\left( \begin{array}{c} \boldsymbol{\beta}^{(0)} \\ \triangle \end{array} \right) \right) \circ \cdots \circ \sigma\left( \phi\left( \begin{array}{c} \boldsymbol{\beta}^{(K-1)} \\ \triangle \end{array} \right) \right) \qquad \text{[IH]}$$

$$= \sigma(\phi(\boldsymbol{\delta})) \qquad \text{[definition]}$$

(iv) weight-preserving:

---

[32]Recall that if $\gamma^{(1)} \in \mathcal{V}$, then $\widetilde{\gamma^{(1)}} = \gamma^{(1)}$.

$$\omega(\boldsymbol{\delta}) = \underbrace{w\Big(\gamma^{(1)} \to \boldsymbol{\beta}^{(0)}\Big)}_{\text{}} \otimes \underbrace{\left(\bigotimes_{i=2}^{K} w\Big(\gamma^{(i)} \to \gamma^{(i-1)}\,\boldsymbol{\beta}^{(i-1)}\Big)\right)}_{\text{}} \otimes \underbrace{\left(\bigotimes_{i=0}^{K-1} \omega\Big(\underset{\triangle}{\boldsymbol{\beta}^{(i)}}\Big)\right)}_{\text{}}$$

equal by constr.     equal by constr.

$$= w\Big(\widetilde{\gamma^{(1)}} \to \boldsymbol{\beta}^{(0)}\Big)$$

$$\otimes \left(\bigotimes_{i=2}^{K} w\Big(\widetilde{\gamma^{(i)}} \to \widetilde{\gamma^{(i-1)}}\,\boldsymbol{\beta}^{(i-1)}\Big)\right)$$

$$\otimes\, w\Big(\gamma^{(K)} \to \widetilde{\gamma^{(K)}}\Big) \underset{\text{by constr.}}{\longleftarrow} \mathbf{1}$$

$$\otimes \left(\bigotimes_{i=0}^{K-1} \omega\Big(\phi\Big(\underset{\triangle}{\boldsymbol{\beta}^{(i)}}\Big)\Big)\right) \underset{\text{equal by IH}}{\longleftarrow}$$

$$= \omega(\phi(\boldsymbol{\delta}))$$

We have used the specific weight settings in Def. 3, and that $\mathbf{1}$ is the $\otimes$-identity element.

**Case** $(\widehat{k} < \infty)$: Here, $\boldsymbol{\delta}$ has the form on the left and $\phi$ transforms into the form on the right:

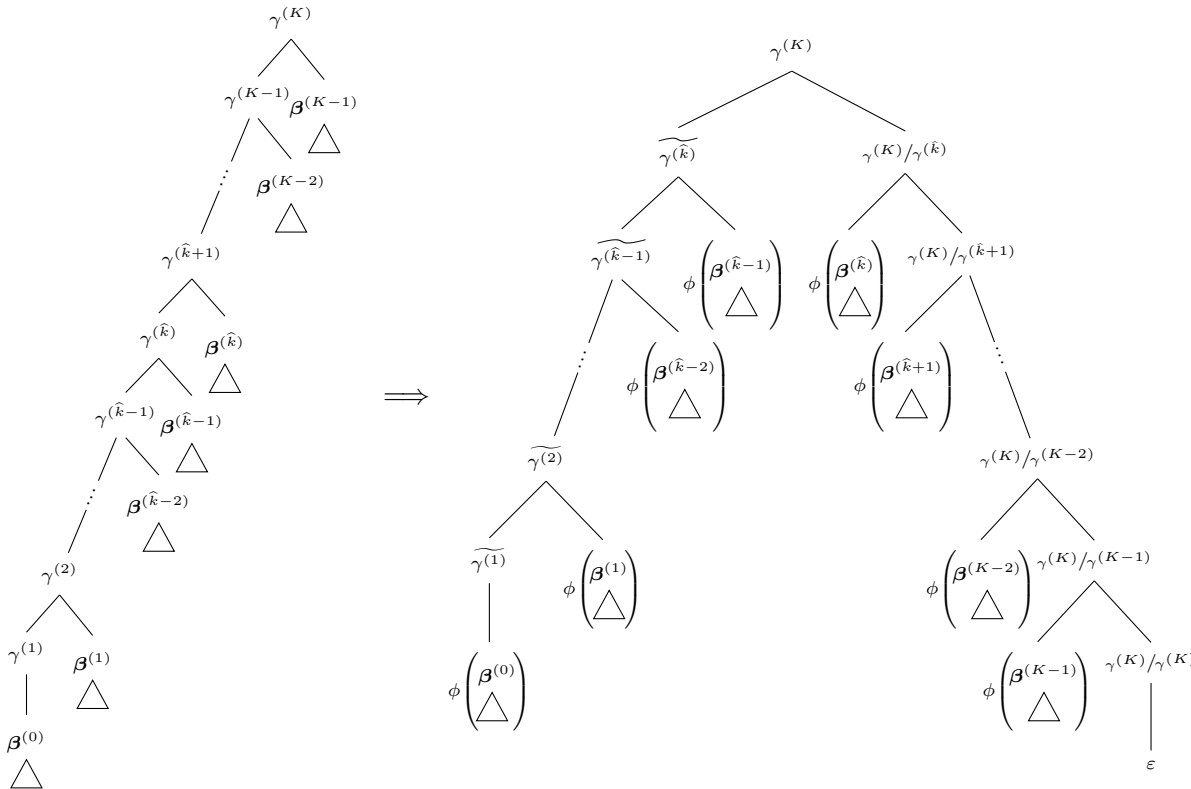

In this case, we observe $\boxed{3}$: $\forall i \in \{1, \ldots, \widehat{k}-1\} : \gamma^{(i)} \notin \overline{\mathcal{C}}$.

We check each of the properties:

(i) $\phi \colon \mathcal{D} \to \mathcal{D}'$: We verify that the rules in the derivation $\phi(\boldsymbol{\delta})$ are in $\mathfrak{G}'$:

- The top rule $\gamma^{(K)} \to \widetilde{\gamma^{(\widehat{k})}}\, \gamma^{(K)}/\gamma^{(\widehat{k})}$ is an instance of (6b). The side condition $\mathrm{X} \in \mathcal{N}, \alpha \in \overline{\mathcal{C}}$ holds because the definition of $\widehat{k}$ ensures that $\gamma^{(\widehat{k})} \in \overline{\mathcal{C}}$.

- The rules in the left subtree follow the same argument as the proof of the $\widehat{k} = \infty$ case.
- Each middle rule of the right subtree $\gamma^{(K)}/\gamma^{(i-1)} \to \boldsymbol{\beta}^{(i-1)} \gamma^{(K)}/\gamma^{(i)}$ (for $i = \widehat{k}{+}1, \dots, K$) is an instance of (6d). The side condition $\mathrm{X} \xrightarrow{w} \mathrm{X}_1 \boldsymbol{\beta} \in \overline{\mathcal{R}}, \mathrm{Y} \in \mathcal{N}$ hold because $\gamma^{(K)} \in \mathcal{N}$, and each of the $\gamma^{(i)} \to \gamma^{(i-1)} \boldsymbol{\beta}^{(i-1)}$ rules was part of the spine (thus, in $\overline{\mathcal{R}}$).
- The bottom rule of the right subtree $\gamma^{(K)}/\gamma^{(K)} \to \varepsilon$ is an instance of (6f).

Lastly, because each $\boldsymbol{\beta}^{(i)}$ subtree is in $\mathcal{D}'$ by IH, we have that $\phi(\boldsymbol{\delta}) \in \mathcal{D}'$.

(ii) **root-preserving:** It is clear by inspection that $\boldsymbol{\delta}$ and $\phi(\boldsymbol{\delta})$ have the same root label.

(iii) **yield-preserving:** The argument here is essentially the same as the $\widehat{k} = \infty$ case. The only difference is that the empty string ($\varepsilon$) will be concatenated as the right-most element, which does not change the yield as $\varepsilon$ is the identity element of the string-concatenation operator.

(iv) **weight-preserving:** The argument is similar to the $\widehat{k} = \infty$ case

$$\omega(\boldsymbol{\delta}) = \underbrace{w\left(\gamma^{(1)} \to \boldsymbol{\beta}^{(0)}\right)}_{\text{equal by constr.}} \otimes \underbrace{\left(\bigotimes_{i=2}^{K} w\left(\gamma^{(i)} \to \gamma^{(i-1)} \boldsymbol{\beta}^{(i-1)}\right)\right)}_{} \otimes \underbrace{\left(\bigotimes_{i=0}^{K-1} \omega\left(\overset{\boldsymbol{\beta}^{(i)}}{\triangle}\right)\right)}_{}$$

$$= w\left(\widetilde{\gamma^{(1)}} \to \boldsymbol{\beta}^{(0)}\right)$$

$$\otimes \left(\bigotimes_{i=2}^{\widehat{k}} w\left(\widetilde{\gamma^{(i)}} \to \widetilde{\gamma^{(i-1)}} \boldsymbol{\beta}^{(i-1)}\right)\right)$$

$$\otimes w\left(\gamma^{(K)} \to \widetilde{\gamma^{(\widehat{k})}} \gamma^{(K)}/\gamma^{(\widehat{k})}\right) \xleftarrow{\text{by constr.}} \mathbf{1}$$

$$\otimes \left(\bigotimes_{i=\widehat{k}+1}^{K} w\left(\gamma^{(K)}/\gamma^{(i-1)} \to \boldsymbol{\beta}^{(i-1)} \gamma^{(K)}/\gamma^{(i)}\right)\right)$$

$$\otimes w(\gamma^{(K)}/\gamma^{(K)} \to \varepsilon) \xleftarrow{\text{by constr.}} \mathbf{1}$$

$$\otimes \left(\bigotimes_{i=0}^{K-1} \omega\left(\phi\left(\overset{\boldsymbol{\beta}^{(i)}}{\triangle}\right)\right)\right)$$

$$= \omega(\phi(\boldsymbol{\delta}))$$

*equal by constr.*

*equal by IH*

We have used the specific weight settings and the fact that $\mathbf{1}$ is the identity element of the $\otimes$-operator.

**Conclusion.** We have successfully verified that the properties (i–iv) hold in each possible case; thus, by the principle of induction, Lemma 3 is true. ∎

**Lemma 4.** *Let $\mathfrak{G} = \langle \mathcal{N}, \mathcal{V}, \mathcal{S}, \mathcal{R} \rangle$, $\mathfrak{G}' = \mathrm{GLCT}(\mathfrak{G}, \overline{\mathcal{R}}, \overline{\mathcal{C}})$ for any choice of $\overline{\mathcal{R}}$ and $\overline{\mathcal{C}}$. Then, the functions $\phi$ and $\phi^{-1}$ defined by Alg. 1 and Alg. 2 (respectively) are inverses of each other. Thus, a bijection of type $\mathcal{D}_{\mathcal{N}} \to \mathcal{D}'_{\mathcal{N}}$ exists.*

*Proof.* **N.B.** We first explain why only a bijection of type $\mathcal{D}_{\mathcal{N}} \to \mathcal{D}'_{\mathcal{N}}$ rather than a bijection of type $\mathcal{D} \to \mathcal{D}'$. The reason is that the set $\mathcal{D}'$ contains derivations rooted at the new slashed and frozen nonterminals, which do not exist in $\mathfrak{G}$. However, all derivations with a root label in $\mathcal{N}$ are preserved in the bijection; that is why we restricted the type of the bijection to have the range $\mathcal{D}'_{\mathcal{N}}$.

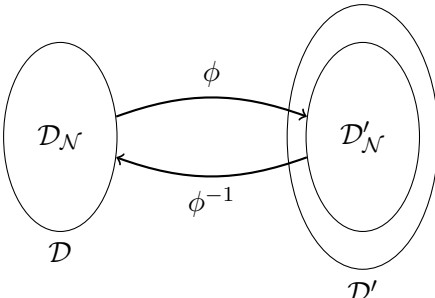

Note that it would be sufficient to show a bijection between $\mathcal{D}_{\mathcal{S}} \to \mathcal{D}'_{\mathcal{S}}$, but we prove the stronger version as it fits with our inductive proof strategy. We note that the derivations in $\mathcal{D}' \setminus \mathcal{D}'_{\mathcal{N}}$ are each labeled by either a frozen or slashed nonterminals symbol. The meanings of these frozen and slashed symbols do not need to be preserved since they are undefined in $\mathfrak{G}$. What is important is that the derivations in $\mathcal{D}'_{\mathcal{N}}$ that use these derivations use them in a meaning-preserving manner.

Our proof will refer to the explicit functions $\phi$ and $\phi^{-1}$ (Alg. 1 and Alg. 2). We will prove that $\phi$ and $\phi^{-1}$ are inverses of each other in two parts:

- *Part 1*: We will show that $\forall \boldsymbol{\delta}' \in \mathcal{D}'_{\mathcal{N}} \colon \phi\big(\phi^{-1}\big(\boldsymbol{\delta}'\big)\big) = \boldsymbol{\delta}'$.

- *Part 2*: We will show that $\forall \boldsymbol{\delta} \in \mathcal{D}_{\mathcal{N}} \colon \phi^{-1}(\phi(\boldsymbol{\delta})) = \boldsymbol{\delta}$.

**Proof of Part 1:** Our proof will proceed by structural induction on the subtree relation ($\prec$). Fix an arbitrary derivation tree $\boldsymbol{\delta}' \in \mathcal{D}'_{\mathcal{N}}$.

**Inductive hypothesis (IH):** For all subtrees $\boldsymbol{\delta}'' \prec \boldsymbol{\delta}'$ where $\ell\big(\boldsymbol{\delta}''\big) \in \mathcal{N} \colon \phi(\phi^{-1}(\boldsymbol{\delta}'')) = \boldsymbol{\delta}''$.

**Base case ($\boldsymbol{\delta}' \in \mathcal{V}$):** $\phi$ and $\phi^{-1}$ are the identity function; thus, $\phi\big(\phi^{-1}\big(\boldsymbol{\delta}'\big)\big) = \boldsymbol{\delta}'$ as required.

**Sequence case:** Both $\phi$ and $\phi^{-1}$ apply pointwise to sequences; thus, $\phi\big(\phi^{-1}\big(\boldsymbol{\delta}'_1 \cdots \boldsymbol{\delta}'_K\big)\big) = \phi\big(\phi^{-1}\big(\boldsymbol{\delta}'_1\big) \cdots \phi^{-1}\big(\boldsymbol{\delta}'_K\big)\big) = \phi\big(\phi^{-1}\big(\boldsymbol{\delta}'_1\big)\big) \cdots \phi\big(\phi^{-1}\big(\boldsymbol{\delta}'_K\big)\big) = \boldsymbol{\delta}'_1 \cdots \boldsymbol{\delta}'_K$ as required.

**Inductive case:** There are two cases to consider: (1) the top rule is an instance of (6b) or (2) the top rule is an instance of (6a). These cases are mutually exclusive, as a difference in the top rule ensures any pair of derivations are unequal. We can see that these cases are exhaustive, as the only way to have a derivation labeled $\mathcal{N}$ that in $\mathfrak{G}'$ is if the top rule of $\boldsymbol{\delta}'$ is an instance of (6b) or (6a).

- **Case (top rule is (6a))**: In this case, $\delta'$ must have the following form:

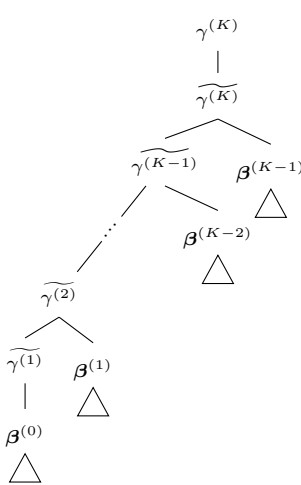

We note that the chain of rules exposed along the left edge must have the following structure: we start with an instance of (6a) followed by zero or more instances of (6f) ending in an instance of (6e). None of the $\beta^{(i)}$-subtrees contain slashed or frozen symbols in their labels. However, note that the *subtrees* of the $\beta^{(i)}$-subtrees may contain slashed or frozen symbols.

We verify that $\phi\big(\phi^{-1}(\delta')\big) = \delta'$ below:

$$\phi\big(\phi^{-1}(\delta')\big) = \phi \left( \vphantom{\begin{array}{c}1\\1\\1\\1\\1\\1\end{array}} \right) = \delta'$$

The first step applies Alg. 2, and the second step applies Alg. 1. By induction, each recursive call in this expression successfully inverts $\phi$ for their respective argument. It is also clear, by inspection, that line 10 of Alg. 1 correctly inverts the transformation of the spine performed on line 9 of Alg. 2. Thus, by induction, our definition of the inverse is correct for this case.

- **Case (top rule is (6b))**: In this case, $\delta'$ must have the following form:

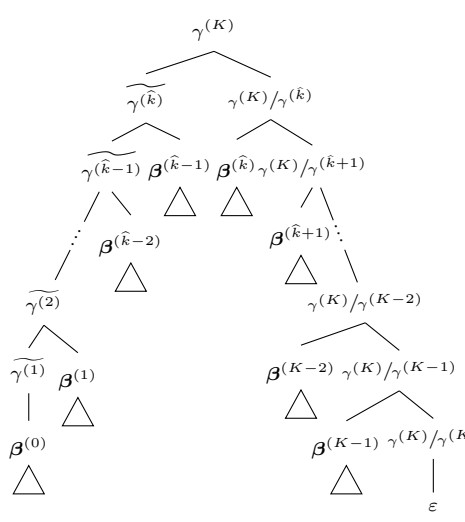

Here, the top rule is an instance of (6b); thus, it combines a derivation labeled by some frozen nonterminal $\gamma^{(\widehat{k})}$ with a derivation of some slashed nonterminal $\gamma^{(K)}/\gamma^{(\widehat{k})}$. Notice that the attached frozen nonterminals $\gamma^{(\widehat{k})}$ must match the slashed nonterminal's denominator, and the slashed nonterminal's numerator must match the label $\ell(\delta')$. The left subtree exposes the chain of rules that built the frozen nonterminal $\gamma^{(\widehat{k})}$. This chain is a (possibly empty) sequence of (6f) followed by the base case (6e). The right subtree exposes the chain of rules that built slashed nonterminal $\gamma^{(K)}/\gamma^{(\widehat{k})}$. These are a (possibly empty) sequence of (6d) ending in an instance of the base case (6c). By construction, none of the $\beta^{(i)}$-subtrees may contain slashed or frozen symbols in their root labels, as they must be instances of one of the rules (6c), (6d), (6e), or (6f).

We verify that $\phi\big(\phi^{-1}(\delta')\big) = \delta'$ by applying Alg. 2 and Alg. 1 in the equations below:

$$\phi\big(\phi^{-1}(\delta')\big)$$

$$= \phi \left( \quad \right)$$

$$= $$

$$= \delta'$$

By induction, each recursive call in this expression successfully inverts $\phi$ for its respective argument. It is also clear, by inspection, that line 12 of Alg. 1 correctly inverts the transformation of the spine performed on line 7 of Alg. 2. Thus, by induction, our definition of the inverse is correct for this case.

**Conclusion (Part 1).** We have verified the two cases of the inductive step; thus, by the principle of induction, we have shown that Part 1 of Lemma 4 holds.

We now continue to Part 2 (next page).

**Proof of Part 2:** Our proof will proceed by structural induction on the subtree relation ($\prec$). Fix an arbitrary derivation tree $\boldsymbol{\delta} \in \mathcal{D}$.

**Inductive hypothesis (IH):** For all subtrees $\boldsymbol{\delta}_1 \prec \boldsymbol{\delta}$: $\phi^{-1}(\phi(\boldsymbol{\delta}_1)) = \boldsymbol{\delta}_1$.

**Base case ($\boldsymbol{\delta} \in \mathcal{V}$):** $\phi$ and $\phi^{-1}$ are the identity function; thus, $\phi^{-1}(\phi(\boldsymbol{\delta})) = \boldsymbol{\delta}$ as required.

**Sequence case:** Both $\phi$ and $\phi^{-1}$ apply pointwise to sequences; thus, $\phi^{-1}(\phi(\boldsymbol{\delta}_1 \cdots \boldsymbol{\delta}_K)) = \phi^{-1}(\phi(\boldsymbol{\delta}_1) \cdots \phi(\boldsymbol{\delta}_K)) = \phi^{-1}(\phi(\boldsymbol{\delta}_1)) \cdots \phi^{-1}(\phi(\boldsymbol{\delta}_K)) = \boldsymbol{\delta}_1 \cdots \boldsymbol{\delta}_K$ as required.

**Inductive case:** Recall from the proof of the inductive case of Lemma 3, $\boldsymbol{\delta}$ must have the general form:

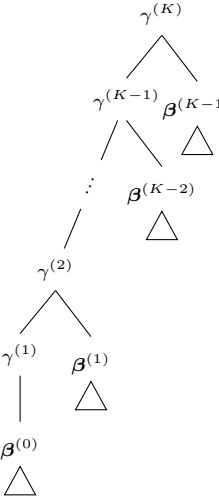

where $K \geq 1$, $\boldsymbol{\beta}^{(i)} \in (\mathcal{V} \cup \mathcal{N})^*, \forall i \{1, \ldots, K\}$, and $(\gamma^{(K)} \to \gamma^{(K-1)} \boldsymbol{\beta}^{(K-1)}) \cdots (\gamma^{(2)} \to \gamma^{(1)} \boldsymbol{\beta}^{(1)})$ is its spine $\zeta(\boldsymbol{\delta})$. (We only show the cases for when $\gamma^{(1)} \in \mathcal{N}$. The case when $\gamma^{(1)} \in \mathcal{V}$ follows the same argument.)

Now, let $\widehat{k}$ be the index of the left corner, or $\infty$ if it does not exist. We consider two cases: $\widehat{k} = \infty$ and $\widehat{k} < \infty$.

**Case ($\widehat{k} = \infty$):** We seek to show that $\phi^{-1}(\phi(\boldsymbol{\delta})) = \boldsymbol{\delta}$ for this case. The equations below depict how spine transformations performed by Alg. 1 and Alg. 2 cancel each other out:

$$\phi^{-1}(\phi(\boldsymbol{\delta})) = \phi^{-1} \left( \begin{array}{c} \gamma^{(K)} \\ | \\ \widetilde{\gamma^{(K)}} \\ \widetilde{\gamma^{(K-1)}} \quad \phi\left(\boldsymbol{\beta}^{(K-1)}\right) \\ \vdots \quad \phi\left(\boldsymbol{\beta}^{(K-2)}\right) \\ \widetilde{\gamma^{(2)}} \\ \widetilde{\gamma^{(1)}} \quad \phi\left(\boldsymbol{\beta}^{(1)}\right) \\ \phi\left(\boldsymbol{\beta}^{(0)}\right) \end{array} \right) = \begin{array}{c} \gamma^{(K)} \\ \gamma^{(K-1)} \quad \phi^{-1}\left(\phi\left(\boldsymbol{\beta}^{(K-1)}\right)\right) \\ \vdots \quad \phi^{-1}\left(\phi\left(\boldsymbol{\beta}^{(K-2)}\right)\right) \\ \gamma^{(2)} \\ \gamma^{(1)} \quad \phi^{-1}\left(\phi\left(\boldsymbol{\beta}^{(1)}\right)\right) \\ \phi^{-1}\left(\phi\left(\boldsymbol{\beta}^{(0)}\right)\right) \end{array} = \boldsymbol{\delta}$$

Thus, by induction, $\phi^{-1}(\phi(\boldsymbol{\delta})) = \boldsymbol{\delta}$ for all derivations $\boldsymbol{\delta}$ that fall into this case.

**Case** $(\widehat{k} < \infty)$: We seek to show that $\phi^{-1}(\phi(\boldsymbol{\delta})) = \boldsymbol{\delta}$ for this case.

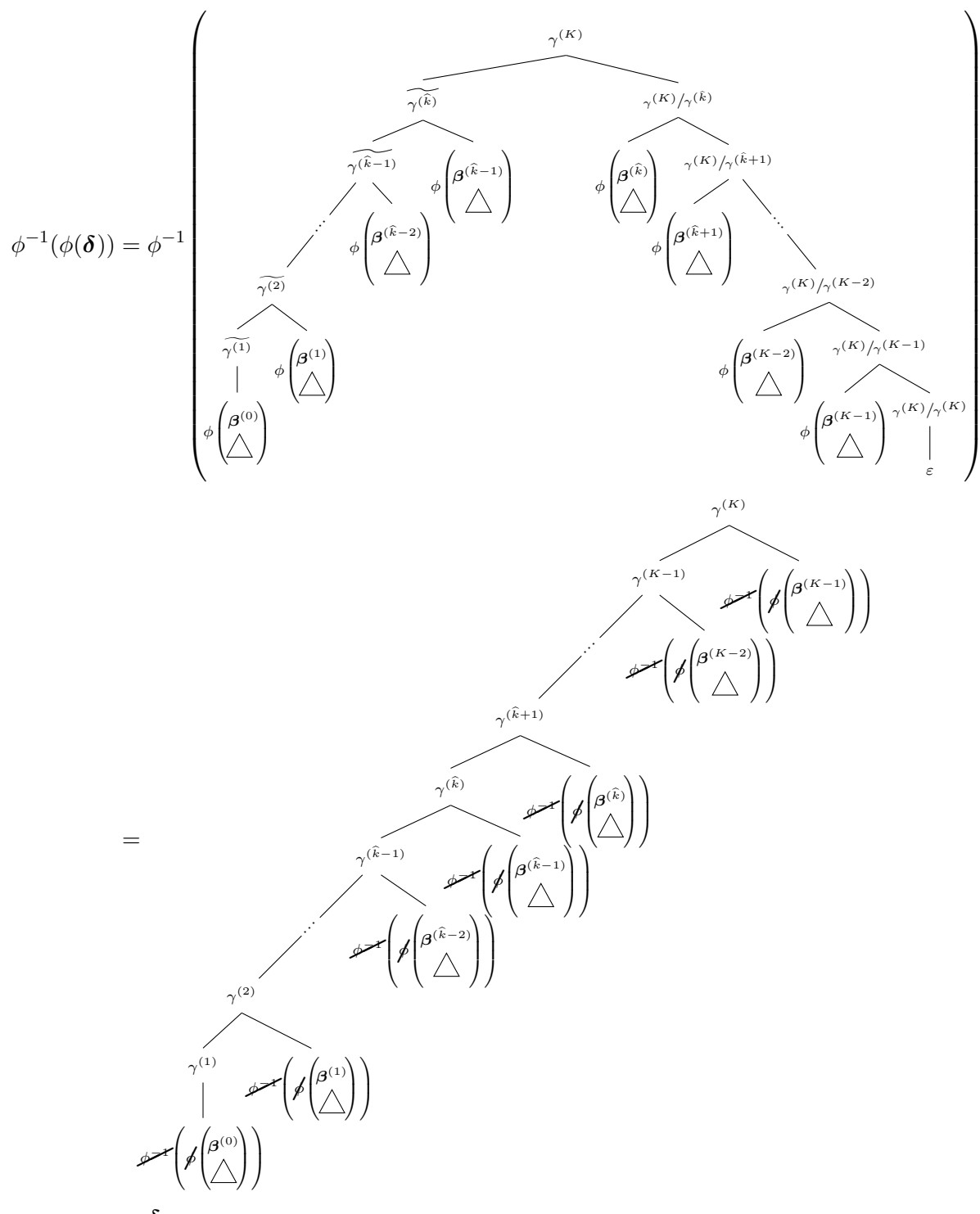

Thus, by induction, $\phi^{-1}(\phi(\boldsymbol{\delta})) = \boldsymbol{\delta}$ for all derivations $\boldsymbol{\delta}$ that fall into this case.

**Conclusion (Part 2).** Since we have shown that each of the possible cases $(\widehat{k} = \infty$ and $\widehat{k} < \infty)$ satisfy $\phi^{-1}(\phi(\boldsymbol{\delta})) = \boldsymbol{\delta}$, it follows that Part 2 of Lemma 4 is true.

**Conclusion.** We have proven Part 1 and Part 2; thus, Lemma 4 holds. ∎

**Theorem 1** ($\mathcal{N}$-bijective equivalence). *Suppose* $\mathfrak{G} = \langle \mathcal{N}, \mathcal{V}, \mathcal{S}, \mathcal{R} \rangle$ *is a WCFG and* $\mathfrak{G}' = \text{GLCT}(\mathfrak{G}, \overline{\mathcal{R}}, \overline{\mathcal{C}})$ *where* $\overline{\mathcal{R}} \subseteq \mathcal{R}$ *and* $\overline{\mathcal{C}} \subseteq \mathcal{V} \cup \mathcal{N}$. *Then,* $\mathfrak{G} \equiv_{\mathcal{N}} \mathfrak{G}'$.

*Proof.* Recall from §2 that $\mathcal{N}$-bijective equivalence ($\mathfrak{G} \equiv_{\mathcal{N}} \mathfrak{G}'$) requires the existence of a structure-preserving bijective mapping of type $\mathcal{D}_{\mathcal{N}} \rightarrow \mathcal{D}'_{\mathcal{N}}$. Lemma 3 shows that $\phi$ in Alg. 1 is a mapping of type $\mathcal{D} \rightarrow \mathcal{D}'$ that preserves the desired structure (label, weight, and yield). Thus, $\phi$ is structure-preserving. Lemma 4 shows that $\phi$ is a bijection of $\phi \colon \mathcal{D}_{\mathcal{N}} \rightarrow \mathcal{D}'_{\mathcal{N}}$. Thus, we have verified the existence of a structure-preserving bijection and, therefore, $\mathfrak{G} \equiv_{\mathcal{N}} \mathfrak{G}'$. ∎

## D   Proof (Sketch) of Theorem 2 (Speculation–GLCT Bijective Equivalence)

**Theorem 2** (Speculation–GLCT bijective equivalence)**.** *For any grammar $\mathfrak{G}$, and choice of $\overline{\mathcal{C}}$ and $\overline{\mathcal{R}}$, $\mathrm{SPEC}(\mathfrak{G}, \overline{\mathcal{R}}, \overline{\mathcal{C}})$ and $\mathrm{GLCT}(\mathfrak{G}, \overline{\mathcal{R}}, \overline{\mathcal{C}})$ are $\mathcal{X}$-bijectively equivalent where $\mathcal{X}$ is the complete set of symbols (i.e., original, frozen, and slashed).*

*Proof (Sketch).* A straightforward inductive argument can be made to show that the derivations of speculation and GLCT under the conditions of Theorem 2 will produce isomorphic trees.

We sketch the derivation mapping from speculation to GLCT. We do not only provide a sketch for the fact that the mapping is a structure-preserving bijection. It is straightforward, albeit laborious, to extend our proof sketch to a proof; such a proof would follow the same structure as our proof of Theorem 1.

**Base case (terminals).**   This case is trivial, as terminals are identical between the transformations.

**Inductive Case.**   We consider three cases for each kind of nonterminal: original, frozen, and slashed. In the diagrams below, we show how the backbone of the derivation is changed. We note that the $\beta$-subtrees of the tree are transformed recursively. In each of the diagrams below, the speculation derivation is on the left, and its corresponding GLCT derivation is on the right.

*Original nonterminals.* This case was discussed in Fig. 2.

*Frozen nonterminals.* The rules defining the frozen nonterminals are identical between the transformations, so the backbone is unchanged in this case.

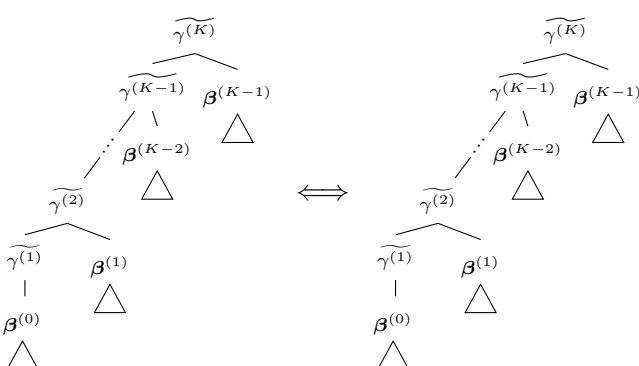

*Slashed nonterminals.* Recall from §3.2 that the difference between the slashed nonterminals in speculation and GLCT is the recursive rule defining slashed nonterminals ((9d) vs. (6d)). The only effect of this difference is that slashed nonterminals in speculation have left-branching derivations, and in GLCT, they have right-branching derivations. The mapping below works for any slashed nonterminal:

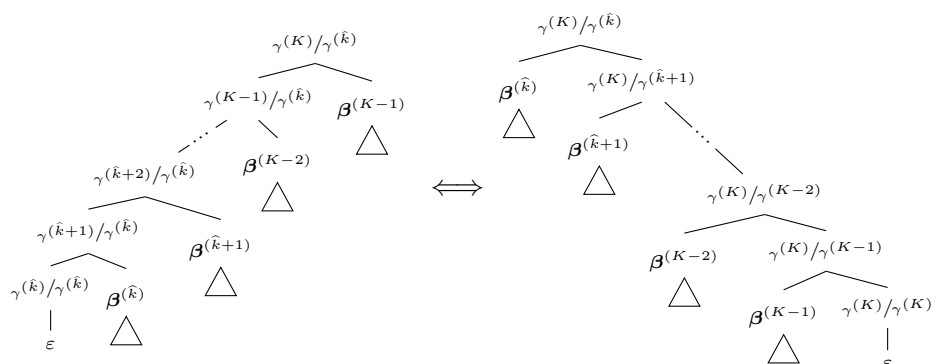

**Structure-preservation and invertibility.** In each of the cases above, it is straightforward to see that mapping is

- label-preserving: the trees in the diagrams each have the same label.
- yield-preserving: the yield of each $\beta$-subtree is preserved (by induction hypothesis), and string concatenation is associative with identity element $\varepsilon$.
- weight-preserving: the weights of rule weights multiplied are equal (by construction), $\otimes$ is associative and commutative (by assumption), and each subtree is equally weighted (by induction hypothesis).
- invertible: the mapping between the trees is invertible because the manipulation of its backbone (i.e., its exposed structure) is invertible, and the mapping for each subtree is invertible (by induction hypothesis).

**Conclusion.** The above cases cover all possibilities. Each sketches the structure-preserving bijective mapping between speculation and GLCT derivations for all their nonterminals (original, slashed, and frozen). Therefore, we have sketched a proof of Theorem 2. ∎

## E   Proof of Theorem 4 (Left-Recursion Elimination)

**Theorem 4** (Left-recursion elimination). *Suppose that* $\mathfrak{G}' = \text{TRIM}(\text{GLCT}(\mathfrak{G}, \overline{\mathcal{R}}, \overline{\mathcal{C}}))$ *where*

- $\mathfrak{G}$ *has no unary rules*
- $\overline{\mathcal{R}} \supseteq$ *the left-recursive rules in* $\mathfrak{G}$
- $\overline{\mathcal{C}} \supseteq \text{bottoms}(\overline{\mathcal{R}})$

*Then,* $\mathfrak{G}'$ *is not left-recursive. Moreover,* $d(\mathfrak{G}') \leq 2 \cdot C$ *where* $C$ *is the number of SCCs in the left-recursion graph for* $\mathfrak{G}$.

*Proof.* Recall from Theorem 1 that there exists a structure-preserving bijection between $\mathcal{D}_{\mathcal{N}}$ and $\mathcal{D}'_{\mathcal{N}}$. Our proof shows how the mapping $\phi$ (Alg. 1) removes (unbounded) left recursion from $\mathfrak{G}$ by analyzing the structure of its transformed trees. Below, we show how $\phi$ takes any derivation $\boldsymbol{\delta} \in \mathcal{D}$ (*left*) and maps a tree $\boldsymbol{\delta}' = \phi(\boldsymbol{\delta})$ where $\boldsymbol{\delta}' \in \mathcal{D}'$ (*right*).

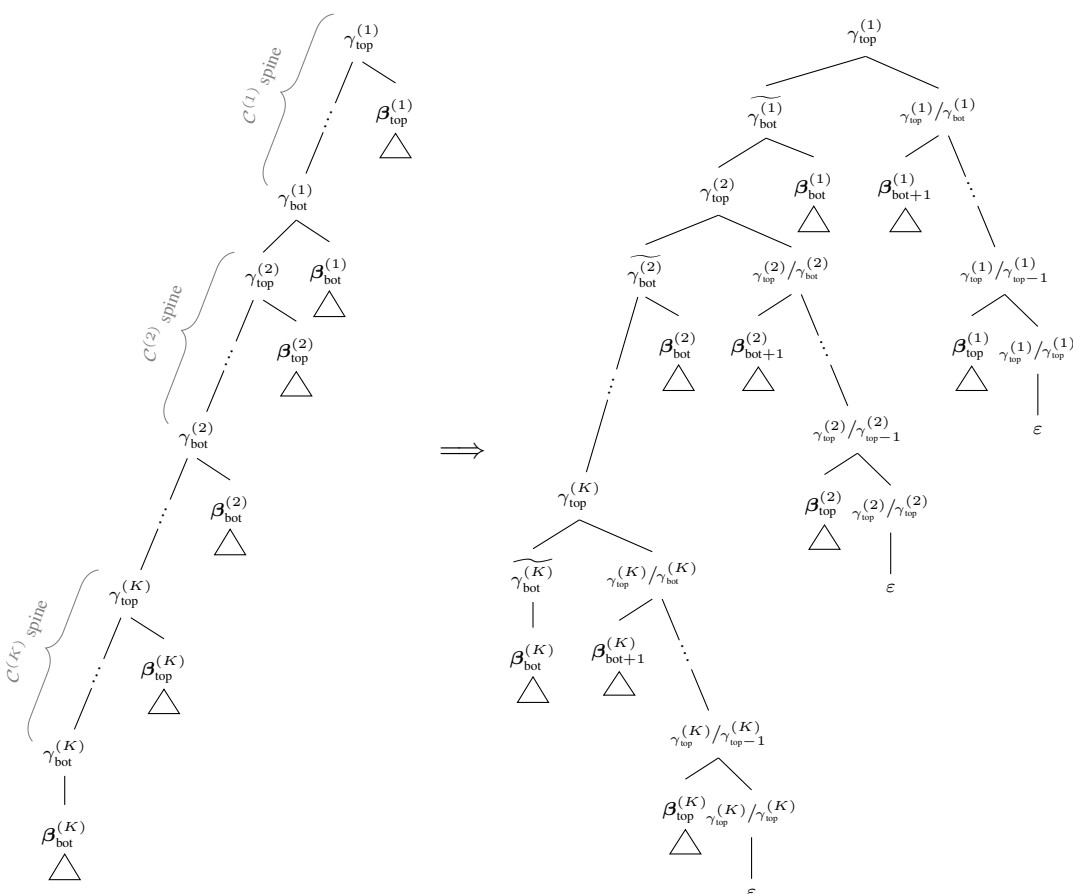

where $\boldsymbol{\beta}_{\text{bot}}^{(K)}$ is a (possibly empty) sequence of terminals. We note that the $\triangle$-subtrees on the right are isomorphic to their counterparts on the left because $\phi$ has (implicitly) transformed them. Let $\mathcal{C}^{(1)}, \mathcal{C}^{(2)}, \ldots, \mathcal{C}^{(K)}$ denote the segmentation of the left edge of $\boldsymbol{\delta}$ into a sequence of spines. The node names indicate its top and bottom elements. Note that spines may contain a single node; $\gamma_{\text{top}}^{(i)} = \gamma_{\text{bot}}^{(i)}$ in these cases. We also note that the transitions between spines occur when a rule on the left is not in $\overline{\mathcal{R}}$.

**Avoiding a subtle mistake.**   We may assume that each $\triangle$-subtree is non-empty because the $\mathfrak{G}$ is unary-free by assumption. This unary-free assumption is important because it ensures that the form of $\boldsymbol{\delta}'$ will not be left-recursive simply because some of its $\triangle$-subtrees are empty.

**The accordion effect.** Recall that $\phi$ (recursively) hoists each left-corner subtree along its respective spine.[33] The accordion effect is the result of repeatedly hoisting: Each spine is contracted into two nodes on the left side of $\boldsymbol{\delta'}$ corresponding to their respective spine's top and bottom elements. Each of the top nonterminals has an unbounded chain of right-recursive rules corresponding to the extracted spine.

The accordion effect assumes that the bottommost node of the spine is always $\overline{\mathcal{C}}$-labeled and, thus, is hoisted by $\phi$ as in the diagram. It is easy to see that our condition on $\overline{\mathcal{C}}$ guarantees this because every node that might end the spine is included in $\overline{\mathcal{C}}$ (by eq. (10)). Furthermore, if $\overline{\mathcal{C}}$ is a superset of $\mathrm{bottoms}(\overline{\mathcal{R}})$, then $\phi$ (by definition) will be unaffected because the bottommost elements of the spines (i.e., those that are hoisted) are already included.[34]

For simplicity, the accordion diagram only illustrates the case where each spine $\mathcal{C}^{(i)}$ is not a singleton containing one element outside of $\overline{\mathcal{C}}$. We now address that case. Here, $\gamma_{\mathrm{top}}^{(i)} = \gamma_{\mathrm{bot}}^{(i)}$ and the rule $\gamma_{\mathrm{top}}^{(i)} \to \gamma_{\mathrm{top}}^{(i+1)}\,\underbrace{\boldsymbol{\beta}_{\mathrm{top}}^{(i)}}$ cannot be in $\overline{\mathcal{R}}$. Like the case depicted, the spine contracts into two nodes on the right: $\gamma_{\mathrm{top}}^{(i)}$ and $\widetilde{\gamma_{\mathrm{top}}^{(i)}}$. However, we now have an instance of (6a) instead of (6b) in the tree on the right; thus, there is no slashed chain of nonterminals.

**Bounding the left-recursion depth.** The result of the accordion effect is that each spine on the left (of unbounded length) is contracted into two nodes in the tree on the right. From here, we can see that $\boldsymbol{\delta'}$ has bounded left-recursive depth as long as there is a bounded number of spines. Recall that each spine is the maximum-length sequence of rules along the left of the derivation starting at the root. Since $\overline{\mathcal{R}}$ includes all left-recursive rules, this means that each spine in the sequence $\mathcal{C}^{(1)}, \mathcal{C}^{(2)}, \ldots, \mathcal{C}^{(K)}$ *completes* at least one SCC—meaning its nodes cannot be visited later in the path. Thus, $d(\boldsymbol{\delta'}) \leq 2 \cdot C$ where $C$ is the number of SCCs in the left-recursion graph $G$ for $\mathfrak{G}$.[35]

**Conclusion.** The above argument bounds $d(\phi(\boldsymbol{\delta}))$ as a function of $\boldsymbol{\delta}$. We can bound $d(\mathfrak{G'})$ by $\max_{\boldsymbol{\delta} \in \mathcal{D}} d(\phi(\boldsymbol{\delta}))$ because $\boldsymbol{\delta}$ and $\boldsymbol{\delta'}$ are in one-to-one correspondence and, thus, no trees in $\mathcal{D}'_{\mathcal{N}}$ will be overlooked. Note that $\mathfrak{G'}$ is trimmed so we can focus on $\mathcal{N}$-derivations. Because our earlier bound, $d(\phi(\boldsymbol{\delta})) \leq 2 \cdot C$, is independent of $\boldsymbol{\delta}$, the maximization is trivial: $d(\mathfrak{G'}) \leq 2 \cdot C$. Thus, the left-recursion in $\mathfrak{G'}$ is bounded, and Theorem 4 holds. ∎

---

[33]This interpretation is discussed throughout the paper: informally in Fig. 1 and §3.1, semi-formally in Fig. 2, and formally in the proof of Theorem 1 (Appendix C).

[34]We note that other ways exist to set $\overline{\mathcal{C}}$ that can break left-recursion cycles. For example, if $\overline{\mathcal{C}}$ is a feedback node set (i.e., a subset of nodes such that removing them and their incoming and outgoing edges results in an acyclic graph). However, the construction is messier, as there will be instances of (6f) along the left of $\boldsymbol{\delta'}$. This still eliminates left recursion, as those frozen nonterminals will not be left recursive. Additionally, it is unclear whether there is any benefit in terms of grammar size or left-recursion depth to these alternatives; thus, we only analyze the simpler case.

[35]This bound can be reduced by including more rules in $\overline{\mathcal{R}}$ at the cost of increasing the size of $\mathfrak{G'}$. For example, if $\overline{\mathcal{R}} = \mathcal{R}$, the left-recursion depth is two, but $\mathfrak{G}$ will likely be larger according to Proposition 3.

## F  Additional Experimental Results

This section presents additional results of the experiments discussed in §4.1. Table 2 complements Table 1 with ratios comparing the size of the output grammars resulting from SLCT with those from GLCT. Appendix F gives analogous results as evaluated on the number of rules. (We note that these results are consistent with those presented in Table 1.)

| Language (size) | Method | Raw | +Trim | $-\varepsilon$'s | Language (size) | Method | Raw | +Trim | $-\varepsilon$'s |
|---|---|---|---|---|---|---|---|---|---|
| Basque | SLCT | 3,354,445 | 245,989 | 411,023 | English | SLCT | 514,338 | 26,655 | 46,088 |
| (73,173) | GLCT | 644,125 | 245,923 | 411,023 | (21,272) | GLCT | 203,664 | 26,289 | 46,088 |
| | SLCT/GLCT | 5.21 | 1 | 1 | | SLCT/GLCT | 2.53 | 1.01 | 1 |
| French | SLCT | 5,902,552 | 106,860 | 173,375 | German | SLCT | 21,204,060 | 272,406 | 434,787 |
| (105,896) | GLCT | 147,860 | 106,628 | 173,375 | (100,346) | GLCT | 1,930,386 | 271,752 | 434,787 |
| | SLCT/GLCT | 39.92 | 1 | 1 | | SLCT/GLCT | 10.98 | 1 | 1 |
| Hebrew | SLCT | 14,304,366 | 564,456 | 979,040 | Hungarian | SLCT | 17,823,603 | 151,373 | 242,360 |
| (84,648) | GLCT | 2,910,538 | 564,074 | 979,040 | (134,461) | GLCT | 748,398 | 151,013 | 242,360 |
| | SLCT/GLCT | 4.91 | 1 | 1 | | SLCT/GLCT | 23.82 | 1 | 1 |
| Korean | SLCT | 54,575,826 | 87,937 | 96,023 | Polish | SLCT | 2,845,430 | 61,333 | 79,341 |
| (59,557) | GLCT | 3,706,444 | 86,529 | 96,023 | (41,957) | GLCT | 177,610 | 61,253 | 79,341 |
| | SLCT/GLCT | 14.72 | 1.02 | 1 | | SLCT/GLCT | 16.02 | 1 | 1 |
| Swedish | SLCT | 20,483,899 | 1,894,346 | 3,551,917 | | | | | |
| (79,137) | GLCT | 6,896,791 | 1,871,789 | 3,551,917 | | | | | |
| | SLCT/GLCT | 2.97 | 1.01 | 1 | | | | | |

Table 2: Same numbers as Table 1 but complemented with the additional SLCT/GLCT rows, which measures the ratio of the SLCT size to the GLCT size.

| Language (# of rules) | Method | Raw | +Trim | $-\varepsilon$'s | Language (# of rules) | Method | Raw | +Trim | $-\varepsilon$'s |
|---|---|---|---|---|---|---|---|---|---|
| Basque | SLCT | 1,060,702 | 62,836 | 145,345 | English | SLCT | 147,221 | 5,941 | 15,653 |
| (28,178) | GLCT | 157,273 | 62,803 | 145,345 | (4,592) | GLCT | 43,724 | 5,758 | 15,653 |
| | SLCT/GLCT | 6.74 | 1 | 1 | | SLCT/GLCT | 3.37 | 1.03 | 1 |
| French | SLCT | 1,970,737 | 31,317 | 64,571 | German | SLCT | 6,891,427 | 71,218 | 152,391 |
| (30,963) | GLCT | 52,545 | 31,201 | 64,571 | (31,156) | GLCT | 466,978 | 70,891 | 152,391 |
| | SLCT/GLCT | 37.51 | 1 | 1 | | SLCT/GLCT | 14.76 | 1 | 1 |
| Hebrew | SLCT | 4,410,783 | 128,186 | 335,456 | Hungarian | SLCT | 5,868,984 | 46,494 | 91,980 |
| (31,587) | GLCT | 612,904 | 127,995 | 335,456 | (43,344) | GLCT | 177,309 | 46,314 | 91,980 |
| | SLCT/GLCT | 7.20 | 1 | 1 | | SLCT/GLCT | 33.10 | 1 | 1 |
| Korean | SLCT | 18,113,579 | 37,342 | 41,382 | Polish | SLCT | 951,176 | 24,105 | 33,104 |
| (28,132) | GLCT | 1,157,353 | 36,638 | 41,382 | (20,109) | GLCT | 61,916 | 24,065 | 33,104 |
| | SLCT/GLCT | 15.65 | 1.02 | 1 | | SLCT/GLCT | 15.36 | 1 | 1 |
| Swedish | SLCT | 5,801,173 | 355,783 | 1,191,574 | | | | | |
| (27,062) | GLCT | 1,272,201 | 351,552 | 1,191,574 | | | | | |
| | SLCT/GLCT | 4.56 | 1.01 | 1 | | | | | |

Table 3: Same settings as in Table 2 but evaluated on the number of rules rather than grammar size.

## G  Filtering Optimization for GLCT

This section shows how the filtering tricks mentioned in §3.1 can be incorporated into the left-corner transformation to reduce the number of useless rules created by Def. 3.[36]

Our first filtering trick is based on Johnson's (1998) strategy for reducing the number of useless rules. It works as follows: the slashed nonterminal $\mathrm{Y}/\mathrm{X}$ is useless if X is not reachable from Y in a left-recursion graph for $\mathfrak{G}$ which only includes the rule set $\overline{\mathcal{R}}$.

Our second filtering trick is based on Moore's (2000) filtering strategy, which extends Johnson's (1998) with additional filtering based on retained nonterminals. The slashed nonterminal $\mathrm{Y}/\mathrm{X}$ is useless if none of the following hold: (i) $\mathrm{Y} = \mathcal{S}$, (ii) Y appears on the right-hand side of some rule in $\overline{\mathcal{R}}$ in a position *other than* the leftmost position, (iii) Y appears on the right-hand side of some rule in $\mathcal{R}\backslash\overline{\mathcal{R}}$. More formally, the set of **retained nonterminals** is

$$R \stackrel{\text{def}}{=} (\{\mathcal{S}\} \cup \{\beta_i \mid (\mathrm{X} \to \alpha\,\boldsymbol{\beta}) \in \overline{\mathcal{R}}, \beta_i \in \boldsymbol{\beta}\} \cup \{\beta_i \mid (\mathrm{X} \to \boldsymbol{\beta}) \in \mathcal{R} \setminus \overline{\mathcal{R}}, \beta_i \in \boldsymbol{\beta}\}) \tag{15}$$

Let $\mathrm{X} \rightsquigarrow \alpha$ denote whether $\alpha$ is reachable from X in the left-recursion graph for $\overline{\mathcal{R}}$. Let $\mathrm{X} \rightsquigarrow \overline{\mathcal{C}}$ denote whether $\exists \alpha \in \overline{\mathcal{C}} : \mathrm{X} \rightsquigarrow \alpha$. Then, we use the following equations for the rules $\mathcal{R}'$:

$$\mathrm{X} \xrightarrow{\mathbf{1}} \widetilde{\mathrm{X}}: \qquad\qquad\qquad\qquad\qquad\qquad\qquad \mathrm{X} \in R\backslash\overline{\mathcal{C}} \tag{16a}$$

$$\mathrm{X} \xrightarrow{\mathbf{1}} \widetilde{\alpha}\, \mathrm{X}/\alpha: \qquad\qquad\qquad\qquad\qquad \mathrm{X} \in R, \mathrm{X} \rightsquigarrow \alpha, \alpha \in \overline{\mathcal{C}} \tag{16b}$$

$$\mathrm{X}/\mathrm{X} \xrightarrow{\mathbf{1}} \varepsilon: \qquad\qquad\qquad\qquad\qquad\qquad \mathrm{X} \in R, \mathrm{X} \rightsquigarrow \overline{\mathcal{C}} \tag{16c}$$

$$\mathrm{Y}/\alpha \xrightarrow{w} \boldsymbol{\beta}\, \mathrm{Y}/\mathrm{X}: \qquad\qquad \mathrm{X} \xrightarrow{w} \alpha\,\boldsymbol{\beta} \in \overline{\mathcal{R}}, \mathrm{Y} \in R, \mathrm{Y} \rightsquigarrow \alpha, \alpha \rightsquigarrow \overline{\mathcal{C}} \tag{16d}$$

$$\widetilde{\mathrm{X}} \xrightarrow{w} \boldsymbol{\alpha}: \qquad\qquad\qquad\qquad\qquad\qquad \mathrm{X} \xrightarrow{w} \boldsymbol{\alpha} \in \mathcal{R}\backslash\overline{\mathcal{R}} \tag{16e}$$

$$\widetilde{\mathrm{X}} \xrightarrow{w} \widetilde{\alpha}\,\boldsymbol{\beta}: \qquad\qquad\qquad\qquad\qquad \mathrm{X} \xrightarrow{w} \alpha\,\boldsymbol{\beta} \in \overline{\mathcal{R}}, \alpha \notin \overline{\mathcal{C}} \tag{16f}$$

Verifying that these filters can only drop useless rules is straightforward.

---

[36] Baars et al. (2010) adopt similar filtering techniques.

## H   Fast Nullary Elimination

Opedal et al. (2023, §F) gives an approach for eliminating nullary rules from a grammar $\mathfrak{G}$ that requires computing the **null weight** $\mathfrak{G}_X(\varepsilon)$ of each nonterminal ($X \in \mathcal{N}$). These null weights are the (smallest) solution to the system of polynomial equations given by (5) where we replace $\mathbf{1}$ with $\mathbf{0}$ in the base case (5a). In general, this system can be solved numerically using fixed-point iteration or Newton's method (see, e.g., Esparza et al., 2007; Nederhof and Satta, 2008; Vieira, 2023). In our case, where $\mathfrak{G}'$ is the output of GLCT, we can solve the system exactly and efficiently, assuming the original grammar $\mathfrak{G}$ is free of nullary rules.[37] Specifically, this system of equations for $\mathfrak{G}'$ is *linear* and can be solved exactly in $\mathcal{O}\big(|\mathcal{N}|^3\big)$ time by using an algorithm for the algebraic path problem (e.g., Lehmann, 1977; Tarjan, 1981) instead of relying on numerical-approximation methods.

**Proposition 4** (Fast null-weight computation)**.** *The null-weight equations for* $\mathfrak{G}' = \text{GLCT}(\mathfrak{G}, \overline{\mathcal{R}}, \overline{\mathcal{C}})$ *are linear, provided* $\mathfrak{G}$ *is nullary free.*

*Proof.* First, we observe that only the nullary rules are those with slashed nonterminal on its left-hand side, $X/X \to \varepsilon$ (6c). We can see that rules created by (6d) and (6b) dictate that these slashed nonterminals only occur on the right-corner position of other rules. It then follows that any derivation with an $\varepsilon$-yield contains at most one nullary rule $X/X \to \varepsilon$ as well as a (possibly zero) number of unary rules on the form $X/Y \to X/z$. This is the case because rules of the form $X/Y \to \boldsymbol{\beta}\, X/z$ with $|\boldsymbol{\beta}| > 0$ contribute a null weight of $\mathbf{0}$ by the assumption that the original grammar has no nullary rules (i.e., each of the original symbols has a null weight of zero, and because GLCT is $\mathcal{N}$-bijectively equivalent, they continue to have a null weight of zero).

The null-weight equations simplify to the following system:

$$\mathfrak{G}'_{X/Y}(\varepsilon) = \bigoplus_{(X/Y \xrightarrow{w} \beta_1 \cdots \beta_K\, X/z) \in \mathcal{R}'} w \otimes \underset{\mathbf{0}}{\mathfrak{G}'_{\beta_1}(\varepsilon)} \otimes \cdots \otimes \underset{\mathbf{0}}{\mathfrak{G}'_{\beta_K}(\varepsilon)} \otimes \mathfrak{G}'_{X/z}(\varepsilon) \quad \oplus \bigoplus_{(X/Y \xrightarrow{w} \varepsilon) \in \mathcal{R}'} w \tag{17}$$

$$= \bigoplus_{(X/Y \xrightarrow{w} X/z) \in \mathcal{R}'} w \otimes \mathfrak{G}'_{X/z}(\varepsilon) \quad \oplus \bigoplus_{(X/Y \xrightarrow{w} \varepsilon) \in \mathcal{R}'} w \tag{18}$$

We can represent this system as a linear equation. We define a matrix $\mathbf{W} \in \mathbb{W}^{\mathcal{N}' \times \mathcal{N}'}$ and vector $\mathbf{v} \in \mathbb{W}^{\mathcal{N}'}$. For each $X/Y, X/z, X/X \in \mathcal{N}'$:

$$\mathbf{W}_{X/Y, X/z} = \bigoplus_{(X/Y \xrightarrow{w} X/z) \in \mathcal{R}'} w \quad \text{and} \quad \mathbf{v}_{X/X} = \bigoplus_{(X/X \xrightarrow{w} \varepsilon) \in \mathcal{R}'} w \tag{19}$$

Then, the null weight for each $\alpha \in \mathcal{N}'$ is

$$\mathfrak{G}'_{\alpha}(\varepsilon) = [\mathbf{W}^* \mathbf{v}]_{\alpha} \tag{20}$$

where $\mathbf{W}^*$ is the solution to $\mathbf{W}^* = \mathbf{I} \oplus \mathbf{W}\mathbf{W}^*$. In the case of the real semiring $\mathbf{W}^* = (\mathbf{I} - \mathbf{W})^{-1}$. For other semirings, it may be computed in $\mathcal{O}\big(|\mathcal{N} \cup \mathcal{V}|^3\big)$ time using an algebraic path solver (e.g., Lehmann, 1977; Tarjan, 1981). ∎

---

[37]Suppose $\mathfrak{G}(\varepsilon) = w$. If $w \neq \mathbf{0}$, we require at least one nullary rule of the form $\mathcal{S} \xrightarrow{w} \varepsilon$. This limited type of rule is straightforward to accommodate in our approach, but we have omitted it from this discussion for simplicity.