# OpenReview forum: "An Exploration of Left-Corner Transformations"
_EMNLP/2023/Conference — EMNLP 2023 Main_

### Official Review · Reviewer_WpEo · 2023-08-05

**Soundness:** 4

**Excitement:**

4: Strong: This paper deepens the understanding of some phenomenon or lowers the barriers to an existing research direction.

**Paper Topic And Main Contributions:**

* This paper introduces a generalization of the original left-corner transformation ("generalized left-corner transform", GLCT) from Rosenkrantz, building on the "speculation" transformation from Eisner & Blatz (2007).
* Proofs are provided that confirm that 1. GLCT indeed removes left-recursion and 2. it produces bijectively equivalent grammars.
* Futhermore, the relation between GLCT and "selective left-corner transformation" (SLCT) from Johnson & Roark is studied, and SLCT is found to be a special case of the GLCT.
* Finally, it is emprically shown that GLCT produces smaller grammars than SLCT.



**Reasons To Accept:**

* This elegant generalization of left-corner parsing (which is more than 50 years old) and its application on other left-corner transform evolutions is without a doubt a very interesting contribution for the parsing, FG and MOL communities.
* The material is presented in an accessible way, with very good knowledge of the related literature.
* Many details for further research are addressed, not only in the "Limitations" section, but throughout the paper, such as, e.g., the transferability to a right-corner scenario

**Reasons To Reject:**

None as per reviewing guidelines.

**Reproducibility:**

5: Could easily reproduce the results.

**Reviewer Confidence:**

3: Pretty sure, but there's a chance I missed something. Although I have a good feel for this area in general, I did not carefully check the paper's details, e.g., the math, experimental design, or novelty.

**Typos Grammar Style And Presentation Improvements:**

* l. 228: There could be a more formal definition of the condition on \vec{\Gamma}.
* l. 432: Regarding the empirical evaluation, it would be interesting to understand if/how the efficiency gains (l. 471) relate to particular grammar properties. This could be verified with grammars extracted from treebanks with different annotation schemes.

---

> ### Author Rebuttal · Authors · 2023-08-29
>
> Thank you for the review and positive feedback. We are glad that you find our work to be an interesting contribution and appreciate that you caught our hints to future research!
>
> > l. 228: There could be a more formal definition of the condition on \vec{\Gamma}.
>
> Great suggestion – we will add that in the final version.
>
> > l. 432: Regarding the empirical evaluation, it would be interesting to understand if/how the efficiency gains (l. 471) relate to particular grammar properties. This could be verified with grammars extracted from treebanks with different annotation schemes.
>
> We agree that an empirical evaluation on how different grammar properties affect the outputs would be an interesting addition. We are going to evaluate and compare these gains on the grammars from the multilingual SPMRL dataset (Seddah et al., 2013, 2014) for the final version.
>
> References that are not already cited in the paper:
>
> * Overview of the SPMRL 2013 Shared Task: A Cross-Framework Evaluation of Parsing Morphologically Rich Languages (Seddah et al., SPMRL 2013)
> * Introducing the SPMRL 2014 Shared Task on Parsing Morphologically-rich Languages (Seddah et al., SPMRL 2014)

---

### Official Review · Reviewer_FNxN · 2023-08-05

**Soundness:** 4

**Excitement:**

4: Strong: This paper deepens the understanding of some phenomenon or lowers the barriers to an existing research direction.

**Paper Topic And Main Contributions:**

The paper proposes a new version of Left-corner transformation for weighted context-free grammars. This transformation is required for using some standard parsing techniques. The authors show that their method combine the ideas behind Selective Left-Corner Transformation and Speculation Transformation but produces smaller grammars than the former and eliminates left recursion in contrast to the latter. They demonstrate the smaller size of the produced grammars on several artificial examples and the ATIS grammar of English.

**Questions For The Authors:**

A) What do you mean by stating that transformation to Greibach normal form requires no left recursion. Do you subsume the weighted context-free grammars, not the usual ones? As I know, the algorithm for usual CFGs starts from a grammar in Chomsky normal form and left recursion does not make a problem for it.

B) Despite your method produces smaller grammars, the difference might be negligible for practical applications. Do you have any performance measurements on ATIS grammar and some standard parsing algorithm? It would be nice to show that the produced speedup is significant, if so.

**Reasons To Accept:**

* A nice mathematical result with both formal and informal justification. The paper is very clearly written.
* The proposed algorithms has better empirical properties than its predecessors. These properties are demonstrated on a dataset of context-free grammars for several typologically diverse languages.

**Reasons To Reject:**

* The scope of the work seems to be slightly different from typical EMNLP paper, it better suits for a mathematical formal language conference. However, the empirical evaluation made in the rebuttal makes the paper appropriate for EMNLP as well.
* From the point of pure mathematics, the results are rather incremental. However, there are still

**Reproducibility:**

N/A: Doesn't apply, since the paper does not include empirical results.

**Reviewer Confidence:**

4: Quite sure. I tried to check the important points carefully. It's unlikely, though conceivable, that I missed something that should affect my ratings.

**Typos Grammar Style And Presentation Improvements:**

A) I prefer notation A\B for "something that can be appended to A to form B". It corresponds to natural rule B -> A A\B, e.g. of Lambek calculus.

---

> ### Author Rebuttal · Authors · 2023-08-29
>
> Thank you for your overall positive review and feedback. We are glad that you appreciate both the theoretical and empirical results and that you found the paper to be well-written.
>
> > The scope of the work seems to be slightly different from typical EMNLP paper, it better suits for a mathematical formal language conference.
>
> Related to this, we acknowledge your comment on our work having a slightly different scope from a typical EMNLP paper. However, we believe that there is great benefit in seeing contributions in regard to theory and methods in the same venue as you see empirical applications of those methods, not the least for the sake of sparking new ideas. For instance, the LCT is known to improve the stack depth of parsers for left- and right-branching structures (Johnson, 1998), which makes them better for beam search decoding.
>
> > The empirical results are mathematical by nature (for a grammar of a certain structure our algorithm produces smaller grammars), not empirical (for a grammar of a certain structure our algorithm produces grammars that are parsed faster). Thus, the absence of really empirical results.
>
> We chose to evaluate on grammar size following in previous work on LCT transformations (Johnson and Roark, 2000; Moore, 2000). Moreover, (nearly) all context-free parsing algorithms run linearly in grammar size. Experiments on grammar size are more replicable because they do not depend on the details of the computer it was run on or the various orthogonal choices of the parsing algorithm’s implementation. However, we will add a running time comparison in the final version (see response above for detail).
>
> > From the point of pure mathematics, the results are rather incremental.
>
> The LCT is an old well-known transformation and is an integral part of the conversion to GNF. Yet, we were able to generalize and improve upon it. This seems notable to us. Furthermore, the prior transformations: LCT, selective LCT, and speculation do not have correctness proofs. We have provided a detailed correctness proof of bijective equivalence (Thm. 2, App. B.2) and for left-recursion elimination (Thm. 1, App. B.1).
>
> > A) What do you mean by stating that transformation to Greibach normal form requires no left recursion. Do you subsume the weighted context-free grammars, not the usual ones? As I know, the algorithm for usual CFGs starts from a grammar in Chomsky normal form and left recursion does not make a problem for it.
>
> GNF requires that the leftmost symbol on the rhs of every production rule is a terminal. Thus, a grammar in GNF is not left-recursive (and even stricter – there are no left-corner relations at all). See Greibach (1965) for the conversion algorithm. You may be referring to an alternative conversion algorithm that goes through CNF (possibly Koch and Blum (1997)?).
>
> > B) Despite your method produces smaller grammars, the difference might be negligible for practical applications. Do you have any performance measurements on ATIS grammar and some standard parsing algorithm? It would be nice to show that the produced speedup is significant, if so.
>
> We agree that even though the grammar factor will be made smaller it is still useful to examine the effects on parsing speed. We plan to add experiments using NLTK’s top-down recursive descent parser (which is sensitive to left-recursion) in the camera-ready version of the paper in order to get more insights on this and strengthen the empirical section of the paper.
>
> References that are not already cited in the paper:
> * Greibach normal form transformation, revisited (Koch and Blum, STACS 1997)

---

### Official Review · Reviewer_tJ8c · 2023-08-06

**Soundness:** 4

**Excitement:**

2: Mediocre: This paper makes marginal contributions (vs non-contemporaneous work), so I would rather not see it in the conference.

**Paper Topic And Main Contributions:**

The paper propose a novel algorithm called Generalized Left-Corner Transformation (GLTC) to transform weighted left-recursive CFGs into weakly equivalent ones without left-recursion (same yield, same weight, but different parse trees). Interestingly the transformation proves to be close to the so-called "speculation" transformation introduced by Eisner and Blatz (2007), with similar notions of added slashed and frozen non-terminals. The paper is rather formal, with a very long proof of equivalence provided in Appendix B (around 14 pages !).  The only empirical part of the paper is the application of the transformation to a large English PCFG, with a comparison in terme of resulting grammar size with an other existing similar algorithm (SLTC), with or without post-processing (trimming and co-trimming) used to remove useless extra non-terminals and productions.

**Reasons To Accept:**

It is always interesting to explore new variants of algorithms, potentially with better performance (in terms of speed or space). This is the case for this algorithm, at least before the post-processing phase. Still,  the gains in terms of number of productions or grammar size are essentially linear on the small toy proposed grammars in Fig. 6
The relationship with the speculation algorithm is also interesting, opening potentially the way to various strategies to slash non-terminals leading to various ways to reorganize parts of derivation trees.

**Reasons To Reject:**

I am not sure that EMNLP be the best place for this paper. It is really a very formal paper, with a very long technical appendix. The empirical part only addresses the application of GLCT on an English grammar with or without the post-processing phases. There is no run-time evaluation of the resulting grammar.
The gains in terms of sizes versus SLCT are real without post-processing, but are just linear (in Figure 6) and essentially disappear on the large English grammar after post-processing (removing useless non-terminals and productions). And of course I believe one wish to run the reduced grammars rather the non-reduced ones
Also, the comparison have been tried only a single grammar, when it would have been relatively easy to apply it of other existing PCFGs (or to PCFGs extracting from existing treebanks)
My final reserve (even if I don't like it) is more general about the usefulness nowadays of such an algorithm, when grammars have essentially been replaced by neural parsing. But the authors rightly motivate the interest for psycho-linguistics


**Reproducibility:**

4: Could mostly reproduce the results, but there may be some variation because of sample variance or minor variations in their interpretation of the protocol or method.

**Reviewer Confidence:**

4: Quite sure. I tried to check the important points carefully. It's unlikely, though conceivable, that I missed something that should affect my ratings.

**Typos Grammar Style And Presentation Improvements:**

Line 147: it would nice to mention \sigma (the yield \sigma)
Line 149: similar with weight w
Figure 6: the definition of grammar size could be precised

---

> ### Author Rebuttal · Authors · 2023-08-29
>
> Thank you for your review and feedback. We appreciate that you see the contributions in exploring new variants of existing algorithms and the new research opportunities that arise from making the connections between LCT and speculation explicit.
>
> > I am not sure that EMNLP be the best place for this paper. It is really a very formal paper, with a very long technical appendix.
>
> Regarding the "long technical appendix": We believe that several *ACL papers could benefit from more formal treatments of their claims, but reading the appendix is by no means necessary for using or welcoming the method proposed in our paper. Besides, the length of the appendix stems mostly from the many diagrams, which we felt were important to include. The proofs themselves do not use particularly sophisticated techniques (they should be accessible to an undergraduate CS student). Previous papers on the LCT and speculation were informal and we felt that the literature deserved a more complete treatment.
>
> > The empirical part only addresses the application of GLCT on an English grammar with or without the post-processing phases. There is no run-time evaluation of the resulting grammar. [...] Also, the comparison have been tried only a single grammar, when it would have been relatively easy to apply it of other existing PCFGs (or to PCFGs extracting from existing treebanks)
>
> Your main concern seems to be that the paper could benefit from more experiments and stronger results. We agree that the paper would be more complete with experiments on more (possibly non-English) grammars and that it would be interesting to see how our method affects parsing runtime. We therefore plan to make the following additions to the final version:
> 1. Add experiments on the SPMRL dataset (Seddah et al., 2013, 2014), which includes grammars in nine different languages.
> 2. Compare runtime on the output grammars using a top-down parser (i.e., a parser that fails in the presence of left-recursion), such as NLTK’s recursive-descent parser.  This comparison will be easy for us to include. Note that the running time complexity of parsing depends linearly on the grammar size so we would expect parsing to run faster following our generalized LCT. However, we agree that it will be useful to include an analysis of how the grammar size affects parsing speed in practice and thus we will include it in the final version.
>
> > The gains in terms of sizes versus SLCT are real without post-processing, but are just linear (in Figure 6) and essentially disappear on the large English grammar after post-processing (removing useless non-terminals and productions). And of course I believe one wish to run the reduced grammars rather the non-reduced ones
>
> Regarding our experimental results, you point out that our method only leads to marginally smaller grammars after post-processing. While true, the post-processing still incurs a computational cost and without them our method is significantly better.
>
> We also note that producing a smaller grammar is not the only contribution of our new transformation. We also unearthed an interesting connection to speculation (Eisner & Blatz, 2007) and have endowed the LC transformation with additional flexibility (beyond that of the selective LC transformation). The fact that it produces smaller grammars is a nice result, but should be viewed as a bonus in terms of contributions. These results also provide insights on how Johnson and Roark’s (2000) transformation, with fewer degrees of freedom, already achieves performance that is hard to improve on (after post-processing), which we think is interesting to note as well.
>
> > My final reserve (even if I don't like it) is more general about the usefulness nowadays of such an algorithm, when grammars have essentially been replaced by neural parsing. But the authors rightly motivate the interest for psycho-linguistics
>
> While it may be true that modern, state-of-the-art parsing is achieved by neural parsers, many of these methods still involve context-free grammars (e.g, Durrett and Klein, 2015; Drozdov et al., 2019; Kitaev, Lu and Klein, 2022). Our work is complementary to these neural parsers. We can for instance imagine a setting where a practitioner who wants to train a neural Earley parser applies our transformation as a preprocessing step in order to eliminate left-recursion (see Opedal et al. (2023) on how left-recursion interacts with Earley’s algorithm). Similarly, if one wishes to train a top-down neural parser, left-recursion should be eliminated.
>
> > Line 147: it would nice to mention \sigma (the yield \sigma) Line 149: similar with weight w Figure 6: the definition of grammar size could be precised
>
> Thank you for spotting these typos!
>
> References that are not already cited in the paper:
> * Overview of the SPMRL 2013 Shared Task: A Cross-Framework Evaluation of Parsing Morphologically Rich Languages (Seddah et al., SPMRL 2013)
> * Introducing the SPMRL 2014 Shared Task on Parsing Morphologically-rich Languages (Seddah et al., SPMRL 2014)
> * Neural CRF Parsing (Durrett & Klein, ACL-IJCNLP 2015)
> * Unsupervised Latent Tree Induction with Deep Inside-Outside Recursive Auto-Encoders (Drozdov et al., NAACL 2019)
> * Learned Incremental Representations for Parsing (Kitaev et al., ACL 2022)

---

### Meta-Review · Area_Chair_ui1J · 2023-09-18

**Recommendation:** 4

**Metareview:**

This paper proposes a generalized Left-Corner Transformation which eliminates left recursion in left-corner transformations yet can produce bijectively equivalent grammars. The authors also provide a detailed correctness of their bijevtive equivalence and prove to be close to the “speculation” transformation of Eisner and Blatz (2007). Empirical comparison with the selective left-corner transformation of the work of Johnson and Roark (2000) show that their approach can reduce the size of output grammar significantly over eight languages.

---

### Decision · Program_Chairs · 2023-10-07

**Decision:**

Accept-Main

**Comment:**

This paper proposes a generalized Left-Corner Transformation which eliminates left recursion in left-corner transformations yet can produce bijectively equivalent grammars. The authors also provide a detailed correctness of their bijevtive equivalence and prove to be close to the “speculation” transformation of Eisner and Blatz (2007). Empirical comparison with the selective left-corner transformation of the work of Johnson and Roark (2000) show that their approach can reduce the size of output grammar significantly over eight languages.